# Statistical aerosol properties associated with fire events from 2002 to 2019 along with a case analysis in 2019 over Australia

Xingchuan Yang[1], Chuanfeng Zhao[1]*, Yikun Yang[1], Xing Yan[1], Hao Fan[1]

1.  State Key Laboratory of Earth Surface Processes and Resource Ecology, and College of Global Change and Earth

System Science, Beijing Normal University, Beijing, China

*Correspondence to*: Chuanfeng Zhao (czhao@bnu.edu.cn)

**Abstract.** Wildfires are an important contributor to atmospheric aerosols in Australia and could significantly affect regional and even global climate. This study investigates the impact of fire events on aerosol properties along with the

long-range transport of biomass burning aerosol over Australia using multi-year measurements from Aerosol Robotic Network (AERONET) at ten sites over Australia, satellite dataset derived from the Moderate Resolution Imaging Spectroradiometer (MODIS) and Cloud-Aerosol Lidar with Orthogonal Polarization (CALIOP), reanalysis data from Modern-Era Retrospective analysis for Research and Applications, Version 2 (MERRA-2), and back-trajectories from the Hybrid Single Particle Lagrangian Integrated Trajectory (HYSPLIT). The fire count, FRP, and AOD showed distinct

and consistent interannual variations with high values during September-February (Biomass Burning period, BB period) and low values during March-August (non-Biomass Burning period, non-BB period) every year. Strong correlation (0.62) was found between fire radiative power (FRP) and aerosol optical depth (AOD) over Australia. Furthermore, the correlation coefficient between AOD and fire count was much higher (0.63-0.85) during October-January than other months (-0.08-0.47). Characteristics of Australian aerosols showed pronounced difference during BB period and

Non-BB period. AOD values significantly increased with fine mode aerosol dominated during BB period, especially in northern and southeastern Australia. Carbonaceous aerosol was the main contributor to total aerosols during BB period, especially in September-December when carbonaceous aerosol contributed the most (30.08-42.91%). Aerosol size distributions showed a bimodal character with both fine and coarse aerosols particle generally increased during BB period. The mega fires during the BB period of 2019/2020 further demonstrated the significant impact of wildfires on

aerosol properties, such as the extreme increase in AOD for most southeastern Australia, the dominance of fine particle aerosols, and the significant increase in carbonaceous and dust aerosols in southeastern and central Australia, respectively. Moreover, smoke was found as the dominant aerosol type detected at heights 2.5-12 km in southeastern Australia in December 2019 and at heights roughly from 6.2 to 12 km in January 2020. In contrast, dust was detected more frequently at heights from 2 to 5 km in November 2019, January, and February 2020. A case study emphasized

that the transport of biomass burning aerosols from wildfire plumes in eastern and southern Australia significantly impacted the aerosol loading, aerosol particle size, and aerosol type of central Australia.

# 1 Introduction

Biomass burning is a major global source of fine carbonaceous aerosols in the form of organic carbon (OC) and black carbon (BC) (Vermote et al., 2009). Biomass burning aerosols are mainly generated by man-made fires such as agricultural burning, deforestation, and biofuels combustion, and natural fires that include lightning-induced wildfires. The global major source regions of biomass burning aerosols are the sub-Saharan Africa, South America, Southeast Asia, North Australia, and the boreal forest in north hemisphere (Ito and Penner, 2004; Mitchell et al., 2006). Abundant aerosols emitted from fires could affect the Earth's climate system by both direct and indirect effects (Jacobson, 2014). Biomass burning aerosols can not only warm the atmosphere and cool the Earth's surface by reducing sunlight through absorption and scattering, but also can modify cloud microphysical properties by acting as cloud condensation nuclei or ice nuclei (Garrett and Zhao, 2006; Wang et al., 2013; Fujii et al., 2015; Zhao and Garrett, 2015; Grandey et al., 2016; Jiang et al., 2018; Zhao et al., 2018; Yang et al., 2019). Moreover, biomass burning aerosols can cause environmental pollution and thus affect public health (Crippa et al., 2016; He et al., 2016; Yang et al., 2016; Yang et al., 2018; Zhu et al.,2018; Yan et al., 2019; Rooney et al., 2020).

The tropical north Australian is dominated by savanna ecosystem with grassland and woodland, while the southeastern Australia is largely vegetated by eucalyptus forest. Meyer et al. (2012) found that approximately 550,000 km$^2$ of tropical and arid savannahs are burned each year in Australia. Australia contributes about 6-8% of global carbon emissions from biomass burning (van der Werf et al., 2006; Meyer et al., 2008). It is also found that biomass burning aerosols are the dominant aerosol type in northern and southeastern Australia during spring and summer, respectively (Mitchell et al., 2013; Yang et al., 2020b). Northern Australia showed a distinct seasonal cycle in aerosol optical depth (AOD) and Ångström exponent (AE) with the highest aerosol loading occurring in spring (burning season) and the highest AE values occurring in August (Mitchell et al., 2013). Mitchell et al. (2006) investigated the characteristics and radiative impact of the smoke aerosol during the Canberra firestorm of January 2003. They found that the single scattering albedo of the aerosols was ~ 0.96 and the daily mean forcing during the week following the firestorm was a cooling of 50 W m$^{-2}$. Additionally, air quality was also significantly affected by smoke from biomass burning over Australia (Meyer et al., 2008; Reisen et al., 2011; He et al., 2016).

During the period September 2019-February 2020, large areas of southeastern Australia were ravaged by deadly wildfire, causing extensive damages to the property and environment. According to the report of Filkov et al. (2020), a total of 5,595,739 and 1,505,004 hectares were burned, 2,475 and 396 houses were destroyed in New South Wales and Victory during the 2019/2020 fire season, which was the most devastating fire season in the states' history. During the wildfire periods, a large amount of aerosols containing smoke, dust, and other burning material were uplifted into the atmosphere, and were even transported around the world, which significantly affected the regional and global atmospheric chemistry, carbon cycle and surface radiation budget. For example, Torres et al. (2020) found that the mega fires in the New South Wales of Australia injected large amount of carbonaceous aerosols in the stratosphere. These wildfires significantly affected local air quality and visibility, and can also be transported to the New Zealand, resulting in degraded air quality in some cities. Ohneiser et al. (2020) found that the smoke from extreme wildfires in southeastern Australia crossed the Pacific Ocean and arrived at Punta Arenas in South America.

There are many studies that have investigated the fire events and their association with enhancing aerosol loading and degrading air quality in Australia (Mitchell et al., 2006; Luhar et al., 2008; Meyer et al., 2008; Bouya and Box, 2011; Mitchell et al., 2013; Mallet et al., 2017; Chen et al., 2019). However, most of these studies are carried out based on observations at specific region/site or during short time period. Long-term statistical analysis about the properties of fire-induced aerosols is highly demanded over a large domain. In addition, the northern and southeastern Australia have frequently suffered from wildfires, especially during the fire season of 2019/2020. It is also highly valuable to fully understand the impacts of the huge fire events occurred in 2019/2020 over the southeastern Australia. In this study,

multi-year aerosol products from ground-based observations (i.e. Aerosol Robotic Network (AERONET)) and satellite-based technique (e.g. Moderate Resolution Imaging Spectroradiometer (MODIS), Cloud-Aerosol Lidar and Infrared Pathfinder Satellite Observation (CALIPSO)) are used to study the impacts of long-term fire events on aerosol properties over Australia. In addition, we further investigated the aerosol properties associated with the huge fire events in 2019/2020 over southeastern Australia, including the impacts on aerosols over central Austria caused by the long-range transport of biomass burning aerosols.

The paper is organized as follows. Section 2 describes the study area, data and method. Section 3 shows the long-term statistical aerosol properties associated with fire events in Australia, aerosol characteristics in Australia during the period in 2019/2020 with huge fire events, along with the aerosol contribution for a case study from long-range transport of biomass burning aerosols. Section 4 summarizes the findings of the study.

## 2 Study area, data and method

### 2.1 Study area

Australia is situated in Southern Hemisphere between the India and South Pacific Oceans. Australian climate varies greatly throughout the eight states and territories and can be divided into six climatic zones (i.e. temperate, grassland, desert, subtropical, tropical and equatorial) based on modified Köppen climate classification system (Fig. 1(a)). The northern Australia lies in the tropical zone with a wet and dry season. The central of Australia is a desert region with high temperature, evaporation, and low amounts of rain during summer. The southern Australia belongs to the temperate zone with hot dry summer and cold winter. In this study, the observations at ten AERONET sites across Australia were adopted for analysis. Table 1 shows the detailed information of site locations and the available data period at each site. Fig.1(b) shows the Kernel density estimation (KDE) of fire spots over Australia from June 2019 to May 2020, and the Gray shade area is the selected domain for vertical profile analysis in this study.

### 2.2 data

#### 2.2.1 Ground-based data

The AERONET is a global aerosol monitoring network based on ground-based sun-sky photometers. It autonomously measures direct-sun irradiance and directional sky radiance and provides long-term and high-quality datasets of aerosol optical, microphysical, and radiative properties (Holben et al., 1998; Yan et al., 2018). The uncertainty of AERONET direct measurement AOD is ±0.01 at visible wavelengths and ±0.02 at near-UV wavelengths (Dubovik et al., 2000). In this study, level 2.0 quality controlled and cloud screened data of the aerosol spectral deconvolution algorithm (SDA) AOD and AE from AERONET Version 3 were used to understand the aerosol loading and aerosol particle size. The retrievals of aerosol microphysical properties such as particle volume size distribution (dV(r)/dlnr) and single scattering albedo (SSA) are used in this study. The detailed retrieval algorithm can be found in Dubovik et al. (2000) and Dubovik et al. (2006), and hence are not reintroduced in this paper. The retrieval errors of dV(r)/dlnr did not exceed 10% in the maxima of the dV(r)/dlnr and may increase up to 35% for the minimum values of the dV(r)/dlnr in the intermediate particle size range ($0.1 \leq r \leq 7$ μm). The retrieval error of dV(r)/dlnr increased significantly for the edges of the particle size interval but did not significantly affect the derivation of the main features of the particle size distribution (Dubovik et al., 2002). Level 1.5 data were used from January 2018 to May 2020 due to the lack of Level 2.0 data at a few sites during the period. In addition, the Canberra site in southeastern Australia ceased to provide data in August 2017, while the neighboring Tumbarumba site began providing observations from July 2019. Due to the similarity of aerosol sources at the two sites, we combined their data to analyze aerosol properties in southeastern Australia.

### 2.2.2 Satellite remote sensing data

The MODIS is a 36-band imaging radiometer onboard the NASA EOS (Earth Observation System) Terra and Aqua platforms and provides globally long-term atmospheric data to monitor the characteristics and dynamics of aerosols (Remer et al., 2005). Sayer et al. (2014) found that the terrain is arid and bright over much of Australia, the MODIS Dark Target (DT) algorithm often retrieves small positive or negative AOD, while the Deep Blue (DB) algorithm can retrieve more available values with 85% of AOD retrievals falling within the expected error envelope in Oceania. Therefore, in this study, the dataset named "Deep_Blue_Aerosol_Optical_Depth_550_Land_Best_Estimate" from Aqua MODIS C61 level2 aerosol product (MYD04_l2) with a spatial resolution of 10 km during the period July 2002-May 2020 was used to investigate the AOD spatio-temporal variations. The MODIS active fire product (MCD14ML) data refer to the active fire hotspots determined using the thermal anomalies at 1 km pixel resolution (Giglio et al., 2016). The fire hotspot count and fire radiative power (FRP) from the MCD14ML data with confidence level greater than 50% from January 2002 to May 2020 were used to explore their relationship with AOD. The CALIPSO was launched in April 2006 equipped with CALIOP (Cloud-Aerosol Lidar with Orthogonal Polarization), and observes global aerosol-cloud vertical distributions at 532 and 1064 nm during both day and night, which provides a possibility to study the impact of clouds and atmospheric aerosols on the Earth's weather, climate, and air quality (Winker et al., 2003). Omar et al. (2013) found that when cloud cleared and extinction quality controlled CALIOP data was compared with AERONET data with AOD less than 1.0, the mean relative difference between the two measurements was 25% of AERONET AOD. Additionally, the CALIPSO AOD showed a good agreement (R=0.65) with AERONET AOD at Lake Argyle in northern Australia. In this study, the Vertical Feature Mask (VFM) data from level 2 profile product (V4.20) were used to obtain the information of aerosol types (e.g., smoke, dust, polluted dust, etc.) over Australia.

### 2.2.3 Reanalysis data

The Modern-Era Retrospective analysis for Research and Applications, Version 2 (MERRA-2) is the NASA's latest global atmospheric reanalysis product, which is produced by the NASA Global Modeling and Assimilation Office (GMAO) (Gelaro et al., 2017). MERRA-2 provides long-term aerosol assimilation with the horizontal resolution of 0.5° × 0.625° on 72 sigma-pressure hybrid layers between the surface and 0.01 hPa from 1980 to the present (Randles et al., 2017). In this study, the monthly MERRA-2 products were used to analyze the spatial and temporal variations of aerosols over Australia from January 2002 to May 2020. ECMWF (European Centre for Medium-Range Weather Forecasts) Reanalysis v5 (ERA-5) (Hersbach and Dee, 2016) is the latest climate reanalysis product developed by the European Centre for Medium-Range Weather Forecasts. ERA-5 provides global hourly and monthly atmospheric data with a spatial resolution of 0.25°×0.25° at 37 pressure levels, which is widely used in various studies (Zhao et al., 2018; Albergel et al., 2018; Wang et al., 2019). ERA-5 adds additional characteristics to ERA-Interim reanalysis, making it even richer in climate information (Albergel et al., 2018). In this study, monthly U-wind, V-wind, and total precipitation from ERA-5 dataset were used for meteorology analysis.

## 2.3 Method

The relationships among AOD, fire count, fire radiative power (FRP), and total precipitation over Australia were investigated by using MODIS AOD, fire hotspot count and fire radiative power data from MCD14ML with confidence level greater than 50%, and total precipitation data from ERA-5. The total, fine, coarse mode AOD, and AE from AERONET, and the AOD products from MODIS and MERRA-2 were utilized to explore their spatio-temporal variations associated with the fire events over Australia. A case of long-range transport of biomass burning aerosols was selected for case study based on the AERONET observations at Birdsville. The 72-h back trajectories for Birdsville site with altitudes of 200 and 500 m above ground level (a.g.l.) were simulated by using HYSPLIT which was driven by the 1° by 1° horizontal resolution meteorological fields from the Global Data Assimilation System (GDAS). Then, the Meteoinfo version2.4.1 (http://meteothink.org/) was employed to cluster the back trajectories by using the Euclidean

distance method. The HYSPLIT back trajectories, the ERA-5 wind fields, the MODIS and CALIOP AOD products were used to investigate the potential aerosol sources, spatial aerosol loading and the vertical features of the aerosol over the central Australia during the case period.

To evaluate the accuracy of MERRA-2 monthly AOD products, the monthly AERONET AOD values were calculated based on the daily measurements. In this study, the quality controlled and cloud screened AOD from AERONET was used to evaluate the performance of MERRA-2 AOD for only cloud-free condition, while MERRA-2 AOD provides the aerosol information in both cloud-free and cloudy condition. Several statistical variables were used in this study for MERRA-2 AOD performance, including the correlation coefficient (R), the RMSE, and relative mean bias (RMB). R was used to analyze the correlation between AERONET and MERRA-2 AOD. The RMSE (Eq. (1)) and RMB (Eq. (2)) were used to evaluate the uncertainty of MERRA-2 monthly AOD products.

$$RMSE = \sqrt{\frac{1}{n}\sum_{i=1}^{n}\left(AOD_{(MERRA-2)i} - AOD_{(AERONET)i}\right)^2} \qquad (1)$$

$$RMB = \frac{1}{n}\sum_{i=1}^{n}|AOD_{(MERRA-2)i}/AOD_{(AERONET)i}| \qquad (2)$$

## 3 Results and discussion

### 3.1 Long-term statistical aerosol properties associated with fire events in Australia

### 3.1.1 Variations and relationships of fire events, precipitation, and aerosol amount

The fire counts and FRP values provide information of the fire frequencies and emissions. The FRP is increasingly used to quantify the regional and global biomass consumption, trace gas and aerosol amount. Figure 2(a) presents the monthly averaged FRP and fire counts detected by MODIS with the confidence level above 50% during the period January 2002 to May 2020 over Australia. The fire counts and FRP both showed significant interannual variations with monthly mean values of 18,014 and 71.26 MW over Australia, respectively. The high FRP values were observed from September to the following February with a maximum value of 104.62 MW in January. The fire counts showed high occurrence during the period September-December with a mean number of 31,006 over Australia. Figure 2(b) shows the monthly variations of MODIS AOD at 550 nm and total precipitation during the period from January 2002 to May 2020 over Australia. It was clear that the MODIS AOD exhibited a distinct interannual variations with high values during the period September-February (0.038-0.063), and low values during the period March-August (0.025-0.034). The temporal AOD variations over Australia were well correlated with that of FRP (R=0.62) for the whole nineteen-year period, while the temporal AOD variations were weakly correlated with that of fire counts (R=0.43). Furthermore, the peak of the monthly mean AODs coincides with the FRP peak in the months of October-January each year (Table.S1). The correlation between AOD and fire counts was much higher (0.63-0.85) during the period of October-January than other months (-0.08-0.47). This was related to the intensive and frequent fires in the tropical savannas of northern Australia and the temperate southern Australia. It was worth mentioning that the frequent dust activities over Central Australia also contributed to the increase in AOD in spring and summer, which may weaken the correlation coefficient between AOD and FRP or fire counts. The total precipitation was found with high values during the period December-March, and low values in other months. In most cases, the peaks of AOD were observed earlier than the peaks of total precipitation during the period January 2002-May 2020, which was well consistent with the fact that high AOD values (including the maxima) occur in the dry season (typically April - November).

### 3.1.2 Aerosol optical depth, size distribution and single scattering albedo

Generally, the seasonality of MODIS AOD and FRP over Australia showed that the high mean AODs coincided

with the FRP during the period September-February every year. Moreover, the high fire counts were recorded during the period of September-December. Dutta et al. (2016) demonstrated that fire events were frequent and severe over Australia during the period September-February. Mitchell et al. (2013) showed that high AOD values (>0.1) were observed at three sites (i.e. Lake Argyle, Jabiru, and Darwin) in northern Australia during the months September-February with maximum values of ~ 0.28 in October. Higher AOD values were also commonly observed during the months September-February in southeastern Australia (Mitchell et al., 2017; Yang et al., 2020b). Previous studies showed that the biomass burning aerosols dominated during this time in the northern and southeastern Australia (Wardoyo et al., 2006; Mitchell et al., 2006; Radhi et al., 2012). Furthermore, the Australian continent was affected by biomass burning aerosols due to the transport of smoke across the entire region (Mitchell et al., 2013; Yang et al., 2020b). Therefore, we defined the periods of September-February as Biomass burning (BB) period, while the periods of March-August as non-Biomass burning (non-BB) period. Figure 3 shows the temporal variations of annual mean AERONET and MODIS AOD during BB period and non-BB period at ten sites. In general, Aqua MODIS DB AOD and AERONET AOD showed the similar annual variations during both the BB period and non-BB period at most sites. However, there were differences in the trends of MODIS AOD and AERONET AOD at a few sites (e.g., Adelaide Site 7, Birdsville, Lake Lefory), which is mainly due to the missing observations of AERONET at those sites. For example, there was a significant difference between the trends in MODIS and AERONET AOD at Birdsville in 2006 and 2013-2014. It was related to the large amount of missing data of AERONET during the periods October-December in 2016 and October 2013-February 2014, which coincided with the periods of dust outbreak at Birdsville. From Figure 3, it can be seen that both MODIS and AERONET observations showed higher AOD values during BB period than non-BB period at ten sites in Australia. The range of the annual averaged AERONET (MODIS) AODs during BB period was 0.067-0.206 (0.023-0.089), while the range during non-BB period was 0.032-0.087 (0.020-0.067). The result indicated that the AOD during BB period contributed dominantly to the annual AOD at ten AERONET sites. The sites in the northern Australia (i.e., Jabiru and Lake Argyle) had higher averaged AOD values both during BB period and non-BB period compared to other sites, which was related to the wildfire activities during BB period and active sea salt production driven by strong winds during non-BB period. Furthermore, the two sites exhibited significant interannual variations during BB period, especially at Lake Argyle. Additionally, a significant increase in AOD was observed during the BB period of 2019/2020 at sites in southeastern Australia, such as Tumbarumba, Fowlers Gap, and Adekaide Site 7, which can be explained by the intensive and frequent fire activities in this region during that period.

Figure 4 shows the spatial distributions of Aqua MODIS DB AOD averaged for all study period (a), non-BB period during 2002-2020 (b), BB period during 2002-2018 (c), and BB period of 2019/20 (d). Overall, MODIS AODs averaged in different periods show similar spatial distributions but differ in their details. High AODs were observed in northern, southeastern, and central Australia, while low AODs were observed in the desert areas of western Australia. The result was in agreement with the earlier findings (Mehta et al., 2016, 2018). The multi-year average MODIS AOD was 0.040, with high AODs observed in eastern Australia (~0.14) and a gradually decreasing trend westwards. During non-BB period, the averaged MODIS AOD values remain low (0.030). The AOD values were less than 0.02 for most of western Australia with the highest value (0.045) observed in northern Australia. As huge fire events occurred in southeastern Australia during the BB period of 2019/2020, we compared the spatial distributions of MODIS AOD during BB period between the periods of 2002-2018 and 2019/2020. The result showed that comparatively higher aerosol loadings were observed during BB period over Australia. The averaged MODIS AODs during BB period of 2002-2018 and 2019/2020 were 0.051 and 0.073, respectively. The high AOD values larger than 0.1 were recorded over northern and southeastern Australia due to biomass burning, while the high AODs in central Australia were mainly due to dust events over the Lake Eyre Basin during BB period. This result also confirmed that the high aerosol loadings during BB period contributed more to the annual AOD. Furthermore, much more AOD values higher than 0.15 were

observed in southeastern Australia during the BB period of 2019/2020, as compared to the BB period of 2002-2018, which was consistent with the observations of AERONET in southeastern Australia.

Figure 5 shows the monthly averaged aerosol volume-size distribution at nine sites over Australia. In general, the aerosol particle size distributions showed a bimodal lognormal pattern with radius smaller than 0.6 μm as fine mode aerosol and with radius larger than 0.6 μm as coarse mode aerosol (Zheng et al., 2017; Zheng et al., 2019). It was clear that the volume sizes with peak volume concentrations for both fine mode and coarse mode aerosols were higher during BB period than during non-BB period at ten sites. In northern Australia, the peak volume concentrations of fine mode aerosols at Jabiru and Lake Argyle were the highest in November and October with values of 0.029 and 0.046 $\mu m^3/\mu m^2$, while the peak volume concentrations of coarse mode aerosols were the highest in December and January with values of 0.041 and 0.032 $\mu m^3/\mu m^2$, respectively (Figs. 5a, 5b). The result indicated that both fine mode and coarse mode volume concentrations significantly increased during BB period. During the BB period, fires cause a temporary reduction in vegetation cover, which can increase biomass burning emissions which are primarily fine aerosol particles. SSA also showed decreasing trends with increase in wavelengths in most months at Jabiru and Lake Argyle in northwestern Australia, especially during the BB period (Figs. 6a, 6b), which showed stronger absorption in the near-infrared bands. Fires also accelerate soil erosion by winds and promote dust emissions (Ravi et al.,2012). The coarse particles such as dust in northern Australia could have been entrained into the biomass burning plume from local soil and also been transported from central Australian deserts (Winton et al., 2016; Yang et al., 2020b). SSA values at Lake Argyle were lower (< 0.90 at 440 nm wavelength) during September-November, suggesting the relative dominance of absorbing aerosols such as dust and black carbon. Furthermore, the sea salt aerosols from ocean would also contribute to the differences in volume size distributions and SSA among various sites, or between BB and non-BB periods. For example, the peak volume concentration values of fine mode (coarse mode) aerosols at Jabiru was lower (higher) than that at Lake Argyle. Moreover, SSA values at Jabiru were higher during BB period than Non-BB period. This was because of Jabiru's closer proximity to the coast, where sea salt aerosols have a greater impact on atmosphere during the wet season (typically from November to April) (Radhi et al., 2012). In western Australia, the peak values of fine mode (coarse mode) aerosol volume concentration at Learmonth and Lake Lefory were the highest in September (October), and the lowest in January (June) (Figs. 5c, 5d). The coarse mode volume concentrations were obviously larger at Learmonth than at Lake Lefory, peaking at 0.02 to 0.03 $\mu m^3/\mu m^2$ in October, compared with ~0.005 to 0.015 $\mu m^3/\mu m^2$ at Lake Lefory. SSA showed an ambiguous wavelength dependence (i.e., increasing or decreasing with wavelengths) at Learmonth due to the presence of aerosol mixture (Fig. 5c). However, the average SSA values were less than 0.90 at 440 nm wavelength during late spring and summer at Learmonth, showing absorbing properties of coarse particles, which was associated with the site's location in the North-Western dust pathway from the Australian interior deserts (e.g. the Gibson Desert and Great Victoria Desert). The average SSA values generally decreased with increasing spectral range at Lake Lefory possibly due to the anthropogenic emissions and biomass burnings (Yang et al., 2020b) (Fig. 5d). In central Australia, there was a significant increase in volume concentrations of coarse mode aerosols during BB period at Birdsville, Fowlers Gap, and Adelaide Site 7, consistent with the presence of dust activities during the period (Figs. 5e, 5f, 5g). The increase in coarse mode aerosols during this period was consistent with the result from SSA at Birdsville and Fowlers Gap, which showed low SSA values (<0.9) at all wavelength during BB period (Figs. 6e, 6f). Meanwhile, increases in volume concentrations of fine mode aerosols were observed at the three sites, which may result from long-range transport of biomass burning aerosols (Yang et al., 2020b). In eastern Australia, the coarse mode aerosols were dominant in almost all seasons at Lucinda (Fig. 5h). The average SSA values generally increased with increasing spectral range with low values (<0.95) at Lucinda (Fig. 6h). However, the volume concentration of fine mode aerosols increased during BB period with peak value of 0.012 $\mu m^3/\mu m^2$ in November. The fine and coarse mode volume concentrations were both higher in December-January at Canberra during BB period (Fig. 5i). The increase in fine mode volume concentrations was attributed to the forest fires in southeastern Australia, while the increase in coarse

mode volume concentrations was mostly related to dust particles from forest fires and transported from central Australian deserts, along with sea salt particles from ocean (McGowan and Clark., 2008; Murphy et al., 2018; Yang et al., 2020b). During the BB period of 2019/2020, the significant impact of the huge fires on aerosol particle size distributions in southeastern Australia was evident. The average SSA values decreased with increasing spectral range at Tumbarumba (Fig. 6i). Furthermore, Significant increase in fine mode aerosol concentrations can be seen at Tumbarumba, which was located in the area with high frequency and intensity of fires, as well as at Fowlers Gap and Birdsville due to the regional transport of biomass burning aerosols. Additionally, a significant increase in coarse mode aerosol concentrations were found at Birdsville. Furthermore, during the BB period of 2019/2020, the peak value of coarse mode aerosol volume concentration at Birdsville was much higher (~0.058 $\mu m^3/\mu m^2$) than the multi-year averaged maximum in January (~0.025 $\mu m^3/\mu m^2$).

### 3.1.3 Emission and optical properties of aerosol species

Figure 7 shows the comparisons between the monthly AERONET observations and MERRA-2 AOD products at nine sites over Australia during the period 2002-2020. The monthly AERONET and MERRA-2 AOD products showed good agreement with correlation coefficients (R) between 0.59 and 0.92 and with small RMSE values (0.02-0.05) at ten sites. Compared to the AERONET observations, the monthly MERRA-2 AODs were slightly overestimated over Australia (RMB: 1.02-1.42) except for the central region, where highly overestimated AODs were found at Fowlers Gap (1.69) and Birdsville (2.12). Overall, the monthly MERRA-2 AOD products showed good performance. Therefore, we employed the monthly MERRA-2 AOD product to determine the contribution from different kinds of aerosols to the AOD over Australia during the period January 2002-May 2020. Figure 8 depicts the monthly averaged carbonaceous, dust, sulfate, and sea salt AODs from MERRA-2, along with the Aqua MODIS DB AOD over Australia. The monthly mean total AODs from MERRA-2 ranged from 0.044 to 0.213, with high values from September to February (i.e., BB period) ranging from 0.097 (February) to 0.130 (November), and low values of 0.054-0.083 in the rest of months (i.e., non-BB period). In contrast, the monthly MODIS AODs were significantly lower than the MERRA-2 total AODs. However, the MODIS AOD showed a good agreement with the MERRA-2 total AOD in temporal variations (R=0.80). Same to the result from MERRA-2, MODIS AODs also showed higher values during BB period (0.038-0.062) than that during non-BB period (0.025-0.034) over Australia. Considering the similar emission sources of organic carbon and black carbon aerosols, carbonaceous aerosol is used in this study to refer to the summation of organic carbon and black carbon from MERRA-2. The contributions of carbonaceous, dust, sulfate, and sea salt aerosols to the total aerosols were 26.24%, 23.38%, 26.36%, and 24.02% over Australia from January 2002 to May 2020, respectively. In general, carbonaceous aerosol was the major contributor to total aerosols during BB period in each year. Carbonaceous aerosol accounted for 30.08-42.91% of total aerosol from September to December, indicating that carbonaceous aerosol from biomass burning dominated the total aerosol loading during this period. In contrast, dust aerosol had greater contribution (23.14-26.88%) to total aerosol during December-March. Sulfate and sea salt aerosols were the dominant (28.25-30.96%) and the second dominant (26.21-28.44%) aerosol type during the months March-July.

Figure 9 presents the spatial distributions of averaged carbonaceous, dust, sulfate, sea salt mass concentrations (a-h) and AODs (i-p) during BB period and non-BB period over Australian continent. Generally, the spatial distribution of carbonaceous mass concentration can be characterized as high in northern and southeastern Australia while low in southwestern Australia. The northwestern Australia had always been the relatively high concentration region of carbonaceous aerosols during both BB period and non-BB period. However, carbonaceous mass concentrations during BB period (1.84 $\mu g/m^3$) were much higher than that during non-BB period (1.07 $\mu g/m^3$), especially in northern and southeastern Australia with concentrations exceeding 4.00 $\mu g/m^3$. This explained the relatively high volume concentrations of fine mode aerosols during BB period at three sites (i.e. Jabiru, Lake Argyle, and Canberra), which are located in regions with frequent fire activities in northern and southeastern Australia. High carbonaceous AODs (>0.06)

were also observed in northern and southeastern Australia during BB period. The carbonaceous AODs decreased significantly during the non-BB period with AOD values less than 0.024 in most of Australia regions. The high sulfate mass concentrations appeared in northwest and southeast coastal areas with mean values exceeding 0.90 μg/m$^3$, while the low sulfate mass concentrations appeared in western Australia with mean values around 0.24 - 0.50 μg/m$^3$. Similarly, the northwest and southeast coastal areas of Australia showed high sulfate AODs compared to the rest of the country. During BB period, the concentrations and AODs of sulfate achieved to high values (above 1.10 μg/m$^3$, above 0.035, respectively) in northwest and southeast coastal areas, which were densely populated urban regions such as Darwin, Sydney, and Melbourne. The central Australia, especially for Lake Eyre Basin, was a relatively high mass concentration region of dust with mean values above 60 μg/m$^3$ during BB period and non-BB period. The dust concentrations declined spatially from the high value center region (i.e. Lake Eyre Basin) to the coastal areas (< 40 μg/m$^3$). Correspondingly, the dust AODs were greater than 0.048 in central Australia and declined a lot in the coastal areas. Additionally, dust concentrations and dust AODs were significantly higher during BB period (24.01 μg/m$^3$; 0.025) than that during non-BB period (16.81 μg/m$^3$; 0.016). The Australian coastal areas, particularly in the northeast, generally had higher sea salt concentrations and AODs than the continental interior region. The sea salt concentrations exceeded 30 μg/m$^3$ over the coastal areas of northern Australia and achieved to the highest during non-BB period (87 μg/m$^3$). The averaged sea salt AODs and concentrations over Australia were higher during BB period (0.024; 13.86 μg/m$^3$) than during non-BB period (0.017; 11.06 μg/m$^3$).

## 3.2 Aerosol properties over Australia during the 2019/2020 mega fire events

Figure 10 shows the monthly FRP and total active fire counts over Australia from Jun 2019 to May 2020. During the period June 2019-May 2020, the fire counts over Australia increased significantly from August (12,440) and peaked in December (67,272). The number of fire counts then gradually decreased from December to March, after which they increased again. The monthly averages of FRP showed a continuous increase from the lowest value in June (37.61 MW) to the highest value in January (135.47 MW), followed by a decrease from February (132.20 MW) to May (35.79 MW). It was worth noting that few fire counts were found in January and February with much higher FRP compared that of other months, which was mostly related to the intense forest fires in southern Australia during summer. Murphy et al. (2019) demonstrated that the fires in temperate southern Australia were less frequent and more severe with very high biomass consumption per fire. In contrast, fires in the tropical savannas of northern Australia were very frequent but less severe with lower biomass consumption per fire. The percentages of active fire counts in different climate zones of Australia from Jun 2019 to May 2020 were analyzed. Results showed that the temperate zone was with the highest number of active fires, accounting for 38.76 % of the total number of fires, followed by the tropical zone (31.95%), grassland (12.39%), subtropical zone (7.42%), and other zones (9.48%).

Monthly spatial distributions of Aqua MODIS DB AOD from Jun 2019 to May 2020 are shown in Figure 11. During the period June 2019-May 2020, the MODIS AOD over Australia was small (0.026) in June, and increased and peaked in December (0.068), then decreased from December to May (0.027). Correspondingly, the monthly averaged MODIS AODs over Australia were with higher magnitudes (0.059-0.068) in November-January. High AOD values that are larger than 0.30 began to appear in eastern New South Wales of Australia in September. Then from October to January, the regions with mean AOD values greater than 0.3 extended from eastern New South Wales of Australia to the eastern and southeastern Australia (e.g. South Australia, Victoria and the Australian Capital Territory). Moreover, the highest AOD values ranging from 2.05 to 3.03 were observed in southeastern Australia during the months November-January, which was most likely caused by the extreme fires in this region. Additionally, high AOD values were also observed in central Australia, which may be related to the dust activities and long-range transport of biomass burning aerosols.

Figures 10 and 11 showed high FRP and AOD values over Australia during the period September 2019-February 2020, indicating the significant impact of fires on aerosols, particularly in southeastern Australia. We further analyzed the daily averaged AOD at 500 nm from AERONET and Aqua MODIS DB AOD during BB period of 2019/2020 at nine sites (Fig.12). As shown, AERONET and MODIS had varying degrees of missing observations at all sites except for Adele Site 7. However, it was clear that AODs at nine sites during BB period of 2019/2020 were higher than the multi-year average values for the same period which are shown in Figure 3. Especially, the AOD values at sites in southeastern Australia such as Tumbarumba, Adele Site 7, Birdsville, and Fowlers Gap showed multiple peaks with AOD values larger than 0.7 after November. Fine mode AOD showed even higher values (0.042-0.175) during the observation period than coarse mode AOD (0.029-0.105) at nine sites, implying the significant increase of fine mode aerosols at these sites during the fire events during BB period of 2019/2020. Particularly, the highest AOD value reached 4.42 with fine mode AOD of 4.40 and coarse mode AOD of 0.02 (not shown in Fig.12 (i)) on 2 January 2020 at Tumbarumba site. The maximum values of total and fine mode AODs were attributed a lot to the fine particles which were generated by the biomass burning. Additionally, the dominance of fine mode aerosols was observed at Birdsville and Fowlers Gap, which was primarily due to the long-range transport of biomass burning aerosols from eastern and southeastern Australia and will be discussed in Section 3.3. Figure 13 shows the occurrence frequency profile of each aerosol type in each month during the period September 2019-February 2020 from CALIPSO observations over the selected domain in southeastern Australia (gray shadow in Fig.1(b)). During this period, polluted dust was abundant at heights roughly from 0.5 to 4 km, with peak occurrence frequency at heights from 0.5 to 2 km. High occurrence frequency of elevated smoke was observed at heights from 2 to 4 km. During the strongest biomass burning period (i.e., December), smoke was the dominant aerosol type detected at heights 2.5-12 km. Further, smoke was the dominant aerosol detected at heights roughly from 6.2 to 12 km in January. The result was consistent with the finding of Ohneiser et al. (2020), who reported that smoke injected over the source regions (i.e. Southeastern Australia) at heights below 10 km widely remained in the troposphere, and then was injected into higher heights (12-20 km) and transported to South America by the eastward advection. In general, the dust occurrence frequency was higher at the heights of 0-2 km and decreased with the increase of height. However, the occurrence frequency of dust increased at heights roughly from 2 to 5 km in November, January, and February. Wagner et al. (2018) indicated that fire radiative energy released by the combustion of the vegetation leads to a significant increase in near-surface wind speed, atmospheric turbulence, and vortices. Moreover, the removal of vegetation during the burning process and the accompanied dehydration and modification of the soil could consequently enhance the dust mobilization and uplift potential, which finally influenced the concentration and the mean size of aerosol particles over the fire region. McGowan and Clark. (2008) showed that dust from the Lake Eyre Basin can potentially affect the southeastern Australia through the southeast dust transport corridor. Therefore, the increase in the occurrence frequency of dust also explained the relatively high coarse mode aerosol volume concentrations at the sites in southeastern Australia (Fig.5), which was a result of the fire-induced dust emissions caused by the pyro-convection during extreme fire events and long range transport of dust from the Lake Eyre Basin. Clean marine aerosol was the dominant type detected below 0.5 km, due to the proximity of this domain to the West Pacific Ocean. It was evident that the peak occurrence frequencies of clean marine decreased during the period December-January, which was also related to the strong fire activities.

Observations from MODIS and AETONET revealed that the aerosol loading over Australia continent, especially in southeastern Australia, significantly increased during the BB period of 2019/2020. Further, the spatial variations of different aerosol species during the BB periods of 2019/2020 and 2002-2018 were investigated based on MERRA-2 dataset (Fig.14). It is evident that the carbonaceous AODs were greater than 0.08 across most of eastern and southeastern Australia, with a maximum value of 0.42 during the BB period of 2019/2020. The BB period of 2002-2018 showed high carbonaceous AOD values in northern Australia, which was consistent of the finding of Yang et al., (2020a). This was related to the biomass burning in savanna regions of northern Australia during the dry season. Large

differences (>0.04) were apparent between the two periods in southeastern and southwestern Australia, which can be attributed to the huge fires in those regions during the BB period of 2019/2020. The dust aerosols exhibited highly similar spatial distributions in aerosol loadings but differ in their magnitudes during the BB periods of 2019/2020 and 2002-2018. The dust AODs were high in the Lake Eyre Basin with many values larger than 0.08, and decreased gradually to the coastal areas during the two periods. The larger differences (>0.02) in dust AOD for the two periods

were also found in the Lake Eyre Basin, while the smaller differences (<0.01) were found in other regions. The result also confirmed the high coarse mode aerosol volume concentration at Birdsville during the BB period of 2019/2020 as discussed in Section 3.1.2. The sulfate and sea salt AODs showed higher values in coastal areas and lower AOD values inland. Meanwhile, the differences of sulfate and sea salt AODs over Australia between the BB period of 2019/2020 and 2002-2018 were small (<0.01).

**3.3 Case study of long-range transport of biomass burning aerosol**

Many studies showed that dust is the major type of atmospheric aerosol over central Australia (Qin and Mitchell, 2009; Mehta et al., 2018; Mukkavilli et al., 2019). However, the central Australia is also affected by the long-range transport of biomass burning aerosols during BB period. During the BB period of 2019/2020, large amounts of smoke plumes produced by southeastern Australian fire were found transported to the South Pacific and even South America

by the prevailing westerly winds (Ohneiser et al., 2020). Meanwhile, those smoke plumes may be transported to the inland and therefore affect the regional aerosol properties over central Australia. From the Sections 3.1 and 3.2, it is evident that the aerosols over central Australia were significantly affected by the fine mode aerosols during the 2019/2020 fire events, implying the potential contribution form fire events. Here, we use a case study to illustrate the long-range transport of biomass burning aerosols, which occurred on 18-26 December 2019.

Figure 15 shows the daily AOD, AE, and aerosol volume-size distributions on 18-26 December 2019 at Birdsville. The mean values of total, fine mode, coarse mode AOD, and AE were 0.382, 0.096 ,0.286 and 1.366 at Birdsville, respectively. Further, higher fine mode AOD values ranging from 0.095 to 0.650 were observed at Birdsville compared to coarse mode AOD (0.026-0.227) on 18 -26 December 2019, which indicated more contribution of AOD from fine mode aerosols during this period. The peak values of fine mode and coarse mode AODs were 0.650 and 0.227 on 21

and 23 December 2019, respectively. Furthermore, the high volume concentrations of fine mode aerosols were observed on 21 and 22 December 2019, while the high volume concentrations of coarse mode aerosols were observed on 23 and 24 December 2019. These results suggested that Birdsville, which is located at the Lake Eyre Basin, was likely significantly affected by the fine mode aerosol transported from a long distance.

We next investigated how the fine mode aerosol was produced and transported to the Birdsville site. Figure 16

shows the spatial distributions of fire spots and wind fields over Australia (the first row), the MODIS AOD at 550 nm and 72 h back trajectories ending at Birdsville (the second row), and the vertical feature mask of aerosols on 18, 21 and 26 December 2019 (the third row). Fire spots were spread across the Australian continent, but few in the Lake Eyre Basin in central Australia, where the Birdsville site is located. The high density of fire spots was observed in the southeastern and eastern Australia, indicating the intensive fire activities in those regions on 18, 21 and 26 December

2019. The easterly and southeasterly winds were observed at Birdsville on these three days, which may bring smoke plums generated by the fires from the eastern and southeastern Australia. The MODIS AOD showed high values (>0.42) in eastern and southeastern Australia on these three days. It is clear that the high AOD values in eastern and southeastern Australia were caused by biomass burning during those days. The back trajectories ending at Birdsville were similar for both atmospheric levels at heights 200 m and 500 m. On 18 December, the air flows originated from

the southeastern Australia. However, the air masses on 21 and 26 December were from the eastern Australia. The MODIS AOD and back trajectories indicated that the rapid increase in AOD at the Birdsville was associated with the

transport of contaminated aerosols from the eastern and southeastern Australia.

Furthermore, the measurements of aerosol subtype profiles from CALIPSO close to the Birdsville site were selected to examine the aerosol transport to Birdsville. On 18 December, the aerosol layer was a mixed of polluted dust and smoke aerosols and reached up to a height of 5 km at Birdsville and surrounding regions (~26°S). The wind fields and back trajectories at Birdsville revealed that the airflows were from southeastern Australia that were regions with intensive fire events. Those airflows brought large amounts of smoke, which got mixed with the dust, resulting in large amounts of polluted dust. This result was also consistent with the observations from AERONET, which show the difference between fine mode aerosol (0.095) and coarse mode aerosol (0.060) was small on 18 December. According the observation of AERONET, the total and fine mode AOD increased on 18-19 December and 20-22 December. On 21 December, the aerosol layer on the west side of the Birdsville site was smoke aerosol at the height of 4-7 km, which could be transported from the eastern Australia by the air flows. The result was consistent with that shown in Fig. 14, which showed high AOD values and high fine mode aerosol volume concentrations on 21 December. On 26 December, the aerosol type on the west side of Birdsville was mixed of smoke and dust aerosol at the height of 0-5 km. Meanwhile, fine mode aerosols dominate at Birdsville under the influence of easterly wind, which were from the regions with fire activities. The result indicated that aerosol types at Birdsville and surrounding regions were still affected by the long-range transport of biomass burning aerosols from eastern Australia. Overall, during the BB period of 2019/2020, the central Australia was significantly affected by the long-range transport of biomass burning aerosols with both AOD values and the fine particle aerosol contribution significantly increased. This result was confirmed by the strong correlation between FRP and AOD over Australia. Moreover, the result also demonstrated that one reason for the strong correlation of AODs among widely separated sites over Australian continent, which were reported by Mitchell et al. (2013, 2017), was the regional transport of biomass burning aerosols, especially during BB period every year.

## 4 Conclusion

In this study, the impact of long-term fire events on aerosol properties in Australia were investigated by using MODIS fire products, ground-based and satellite-based aerosol products. Further, the spatiotemporal variations of aerosol properties during the 2019/2020 mega fire events and a case study of long-range transport of biomass burning aerosols over Australia were investigated. The main findings are as follows.

1. The fire count, FRP and MODIS AOD exhibited distinct but consistent interannual variations with high values during the period September-February (BB period) every year. The correlations between fire count, FRP and AOD were 0.43 and 0.62 over Australian continent, respectively. Furthermore, Strong correlation between AOD and fire count was found much higher (0.63-0.85) during October-January than other months (-0.08-0.47). The multi-year (2002-2020) averaged MODIS AOD over Australia was 0.040 with high values in eastern Australia (~0.14) and gradually decreased westwards. MODIS AOD showed higher values during BB period (0.051) than during non-BB period (0.030) over Australia. The multi-year average contributions of carbonaceous, dust, sulfate, and sea salt aerosols to the total were 26.24%, 23.38%, 26.36%, and 24.02% over Australia, respectively. Carbonaceous aerosol was the major contributor to total aerosol during BB period, especially during the months September-December when carbonaceous aerosol contributed the most (30.08-42.91%). Result indicated that the AOD during BB period contributed dominantly to the annual AOD with both fine and coarse mode aerosols increased significantly during BB period.

2. During the BB period of 2019/2020, the number of fire count and FRP over Australia significantly increased from September and peaked in December and January, respectively. The mega fires significantly affected the aerosol properties in Australia, especially in southeastern Australia. The total AOD and fine mode AOD values significantly increased with extreme multi-peak values (e.g., 4.42 and 4.40 at Tumbarumba on 2 January 2020, respectively). During the months September -January, the regions with MODIS AOD values greater than 0.3 extended to eastern

and southeastern Australia. Both fine and coarse aerosol particle generally increased in this period. Polluted dust was detected more frequently from 0.5 to 4 km. However, smoke was the dominant aerosol type detected at 2.5-12 km in December 2019 and the dominant aerosol type detected roughly from 6.2 to 12 km in January 2020, while dust was detected more frequently from 2 to 5 km in November 2019, January, and February 2020. Carbonaceous aerosols increased significantly in eastern and southeastern Australia, while dust aerosol increased in central

Australia.

3. A case analysis emphasized the long-range transport of biomass burning aerosols from wildfire plumes in southeastern and eastern Australia significantly impacted central Australian desert regions with a significant increase in AOD and fine mode aerosol contribution. The result demonstrated that an important reason for the strong correlations among fire count, FRP and AOD, and among AODs at widely separated sites over Australian

continent, was the regional transport of biomass burning aerosols, especially during BB period every year.

There are still some limitations in this study. First, ten ground-based AERONET sites may not be sufficient to fully reveal the impact of wildfires on aerosol over Australia, which could have affected the statistics to some extent. Second, we distinguished and defined the BB period and Non-BB period from the Australia continent

perspective based on high and low values of AOD and FRP, which could help to understand the impact of wildfires on aerosols in Australia. However, the high fire frequencies occur during the dry season (generally April to November) in northern Australia, while the high fire frequencies occur in the austral spring and summer month (September-February) in southern and eastern Australia. Hence, future work will consider the difference of fire seasons in Australia continent.

**Data Availability.**

The AERONET dataset were obtained from https://aeronet.gsfc.nasa.gov/. Aqua MODIS AOD data were available from the Atmosphere Archive and Distribution System Distributed Active Archive Center (LAADS DAAC; https://ladsweb.modaps.eosdis.nasa.gov/). HYSPLIT data are provided by the NOAA READY website (http://www.ready.noaa.gov). MERRA-2 Reanalysis data were provided by the NASA Global Modeling and

525 Assimilation Office (https://gmao.gsfc.nasa.gov/reanalysis/MERRA-2/). ERA-5 Reanalysis data were provided by the European Centre for Medium Weather Forecasts (https://cds.climate.copernicus.eu/). CALIPSO data were acquired from the NASA's website (https://www-calipso.larc.nasa.gov/). The classified climate zones of Australia were obtained from Bureau of Meteorology of Australian Government (http://media.bom.gov.au).

**Acknowledgements.**

This research was supported by the National Natural Science Foundation of China (Grants 41925022), the China National Key R&D Program (2019YFA0606803), and the State Key Laboratory of Earth Surface Processes and Resource Ecology. The authors would like to thank AERONET team, NASA Goddard Space Flight Center (GSFC), NASA Global Modeling and Assimilation Office (GMAO), and CALIPSO team for providing aerosol optical properties data. We also thank the LANCE-MODIS Archives and European Centre for Medium-Range Weather Forecasts team for

processing and distributing the active fire and ERA-5 data, respectively.

**Author contributions.**

CFZ designed the research, and CFZ and XCY carried out the research and wrote the manuscript. YKY contributed to collect and analyses CALIPSO aerosol data. HF and XY provided constructive comments on this research. All authors made substantial contributions to this work.

**Competing interests.**

The authors declare that they have no conflict of interest.

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

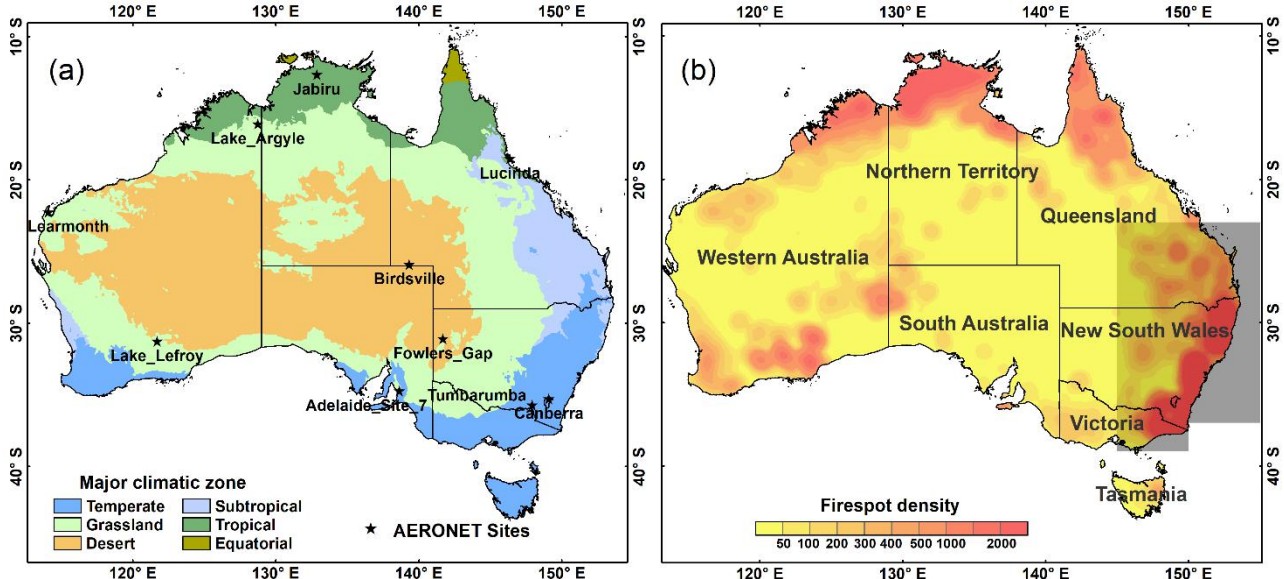

**Figure 1. (a) Climatic zones of Australia defined by Köppen climate classification system. The black stars represent ten AERONET sites located in Australia. The classified climate zones of Australia were obtained from Bureau of Meteorology of Australian Government ([http://media.bom.gov.au](http://media.bom.gov.au)). (b) Kernel density estimation (KDE) of the number of wildfire events based on MCD14DL with confidence level greater than 50% over Australia from Jun 2019 to May 2020. The Gray shade area is the selected domain for vertical profile analysis in this study.**

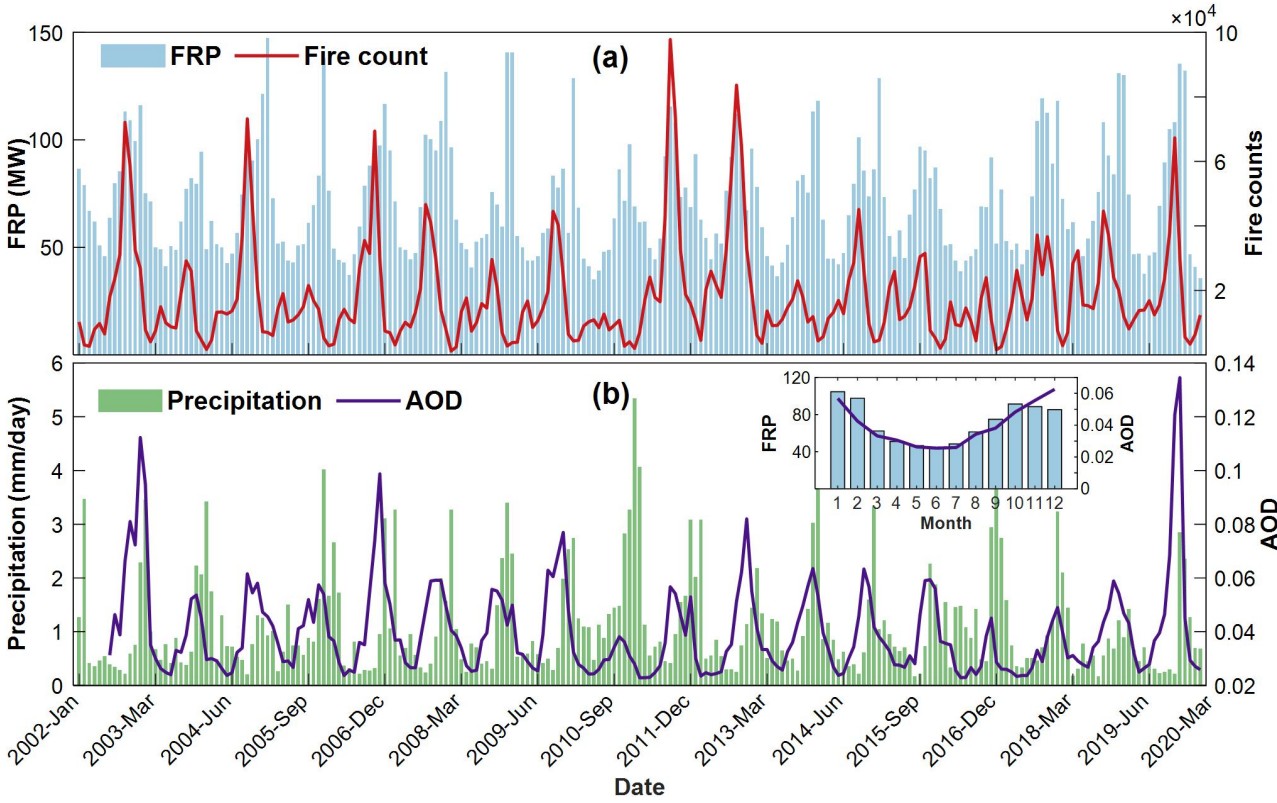

**Figure 2. Time series of monthly averaged fire radiative power (FRP), fire count from MCD14ML (a), and total precipitation from ERA-5 and Aqua MODIS DB AOD (b) in Australia.**

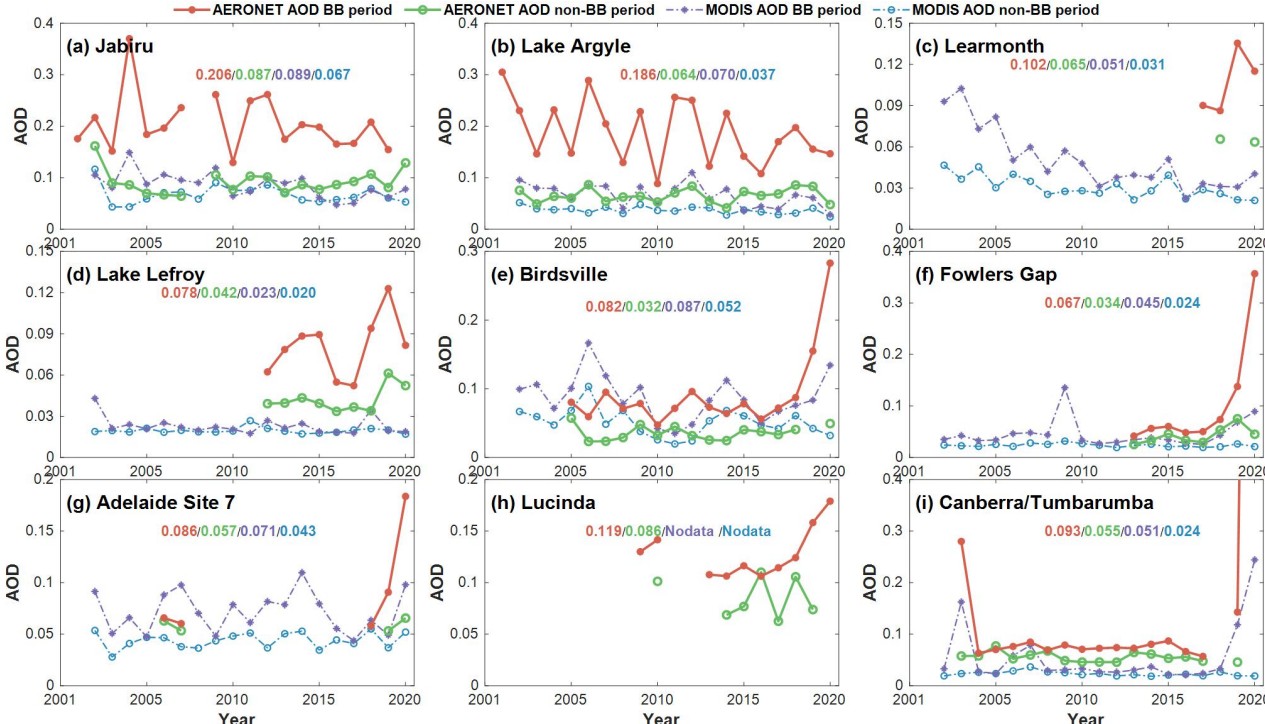

**Figure 3. Temporal variations of annual mean AERONET AOD at 500 nm and Aqua MODIS DB AOD at 550 nm during BB period and non-BB period at nine sites in Australia. The number above with different colors refers to the corresponding average values.**

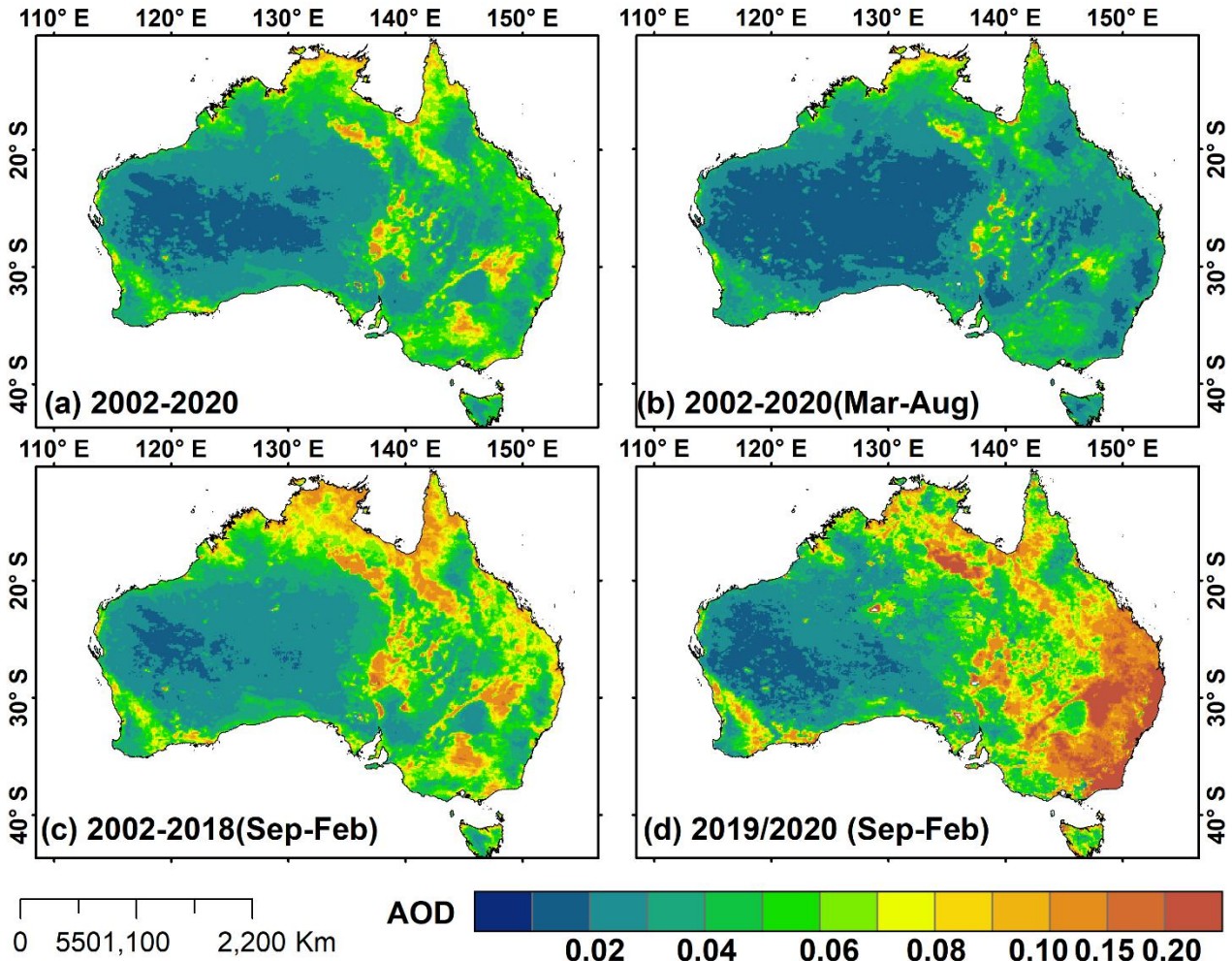

**Figure 4. Spatial distributions of Aqua MODIS DB AOD at 550 nm over Australia during the different period 2002-2020 in Australia, a) July 2002-May 2020, b) March-August (non-BB period) of 2002-2020, c) September-February (BB period) of 2002-2018, and d) BB period of 2019/2020.**

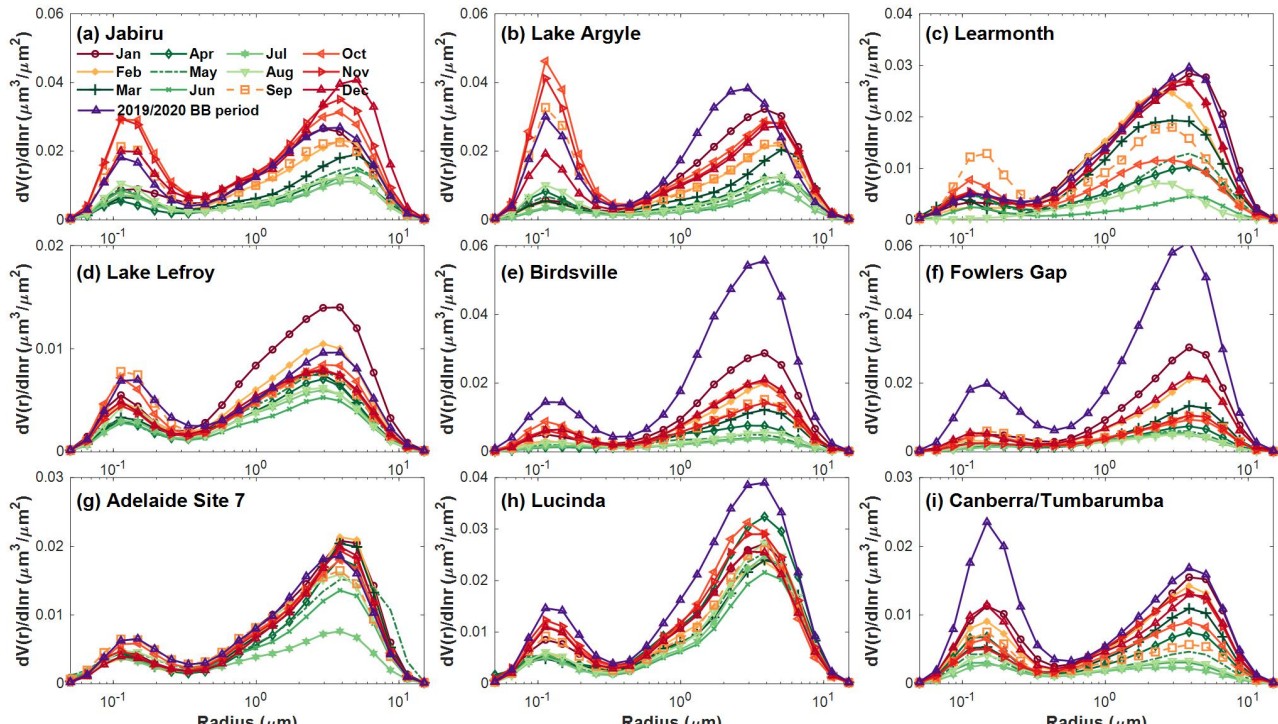

Figure 5. Monthly averaged aerosol volume size distributions at the nine AERONET sites over Australia. The warm-toned and cold-toned lines represent the aerosol size distributions in BB period and non-BB period, respectively. The purple line represents the aerosol size distribution in BB period of 2019/2020.

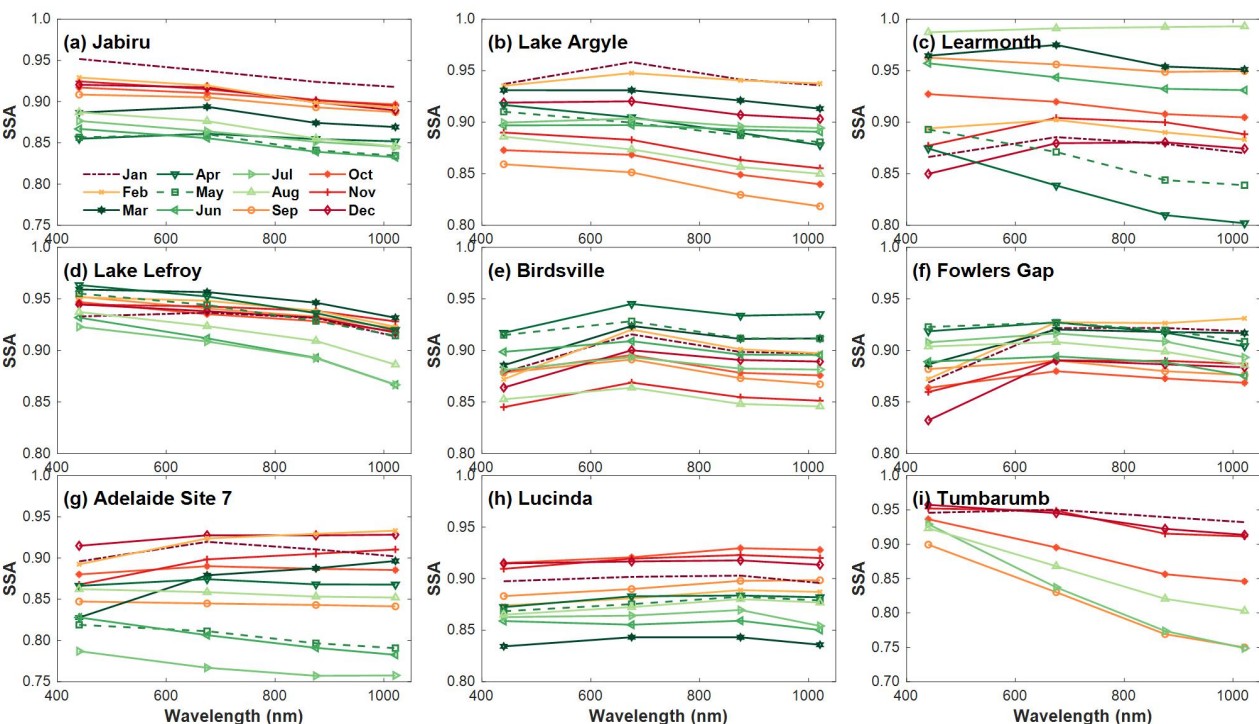

Figure 6. Monthly averaged single scattering albedo (SSA) at the nine AERONET sites over Australia. The warm-toned and cold-toned lines represent the aerosol size distributions in BB period and non-BB period, respectively. Note: Only SSA data from Tunbarumb is used due to the lack of SSA data at Canberra.

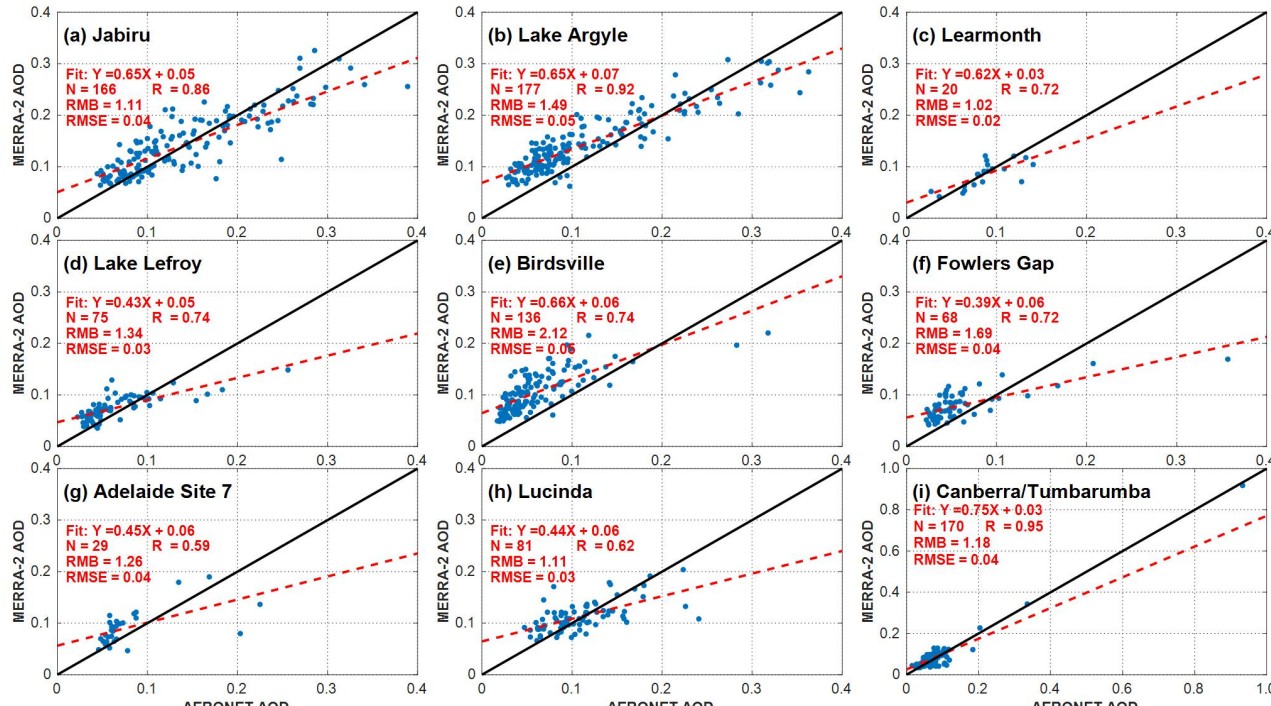

Figure 7. The comparisons of monthly AODs between MERRA-2 product and AERONET at nine AERONET sites in Australia during 2002–2020. (Linear regression is shown as a dashed red line and all the linear relationships are statistically significant at α = 0.01. The black solid line is the 1:1 line.)

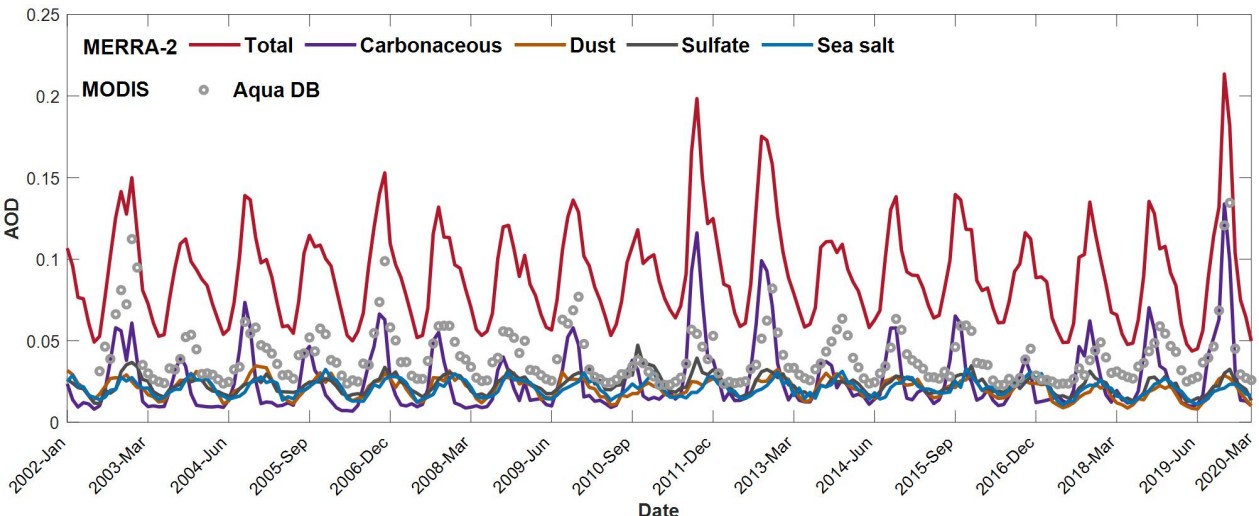

Figure 8. Time series of monthly averaged carbonaceous, dust, sulfate, and sea salt AODs from MERRA-2, along with the Aqua MODIS DB AOD in Australia from January 2002 to May 2020.

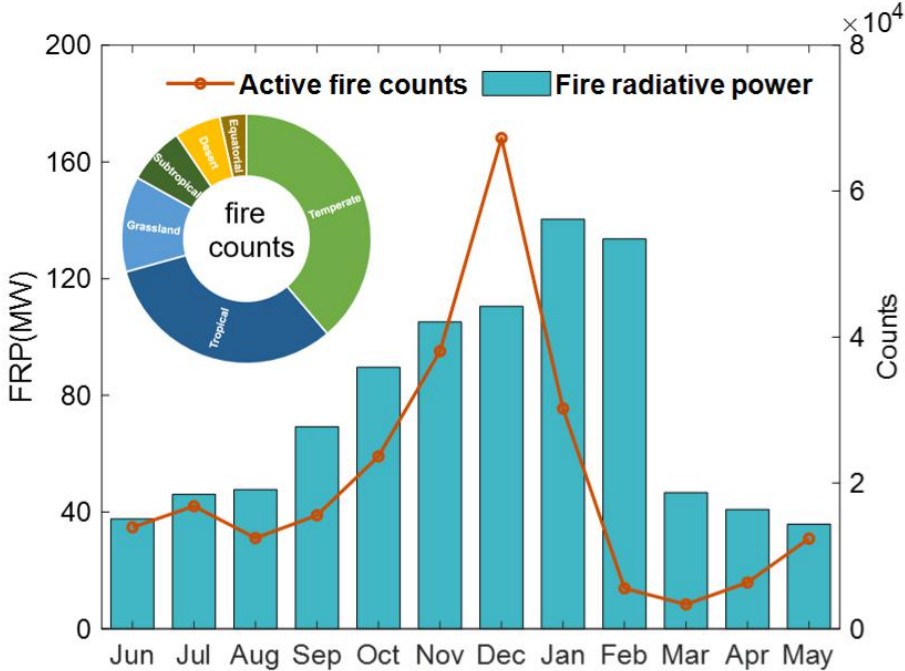

**Figure 9.** Spatial distribution of carbonaceous, dust, sulfate, and sea salt mass concentrations near surface (a-h) and AOD (i-p) estimated by MERRA-2 during BB period and non-BB period over Australia.

**Figure 10. Monthly FRP and total active fire counts in Australia from Jun 2019 to May 2020. The percentages of active fire detections in different climate zones of Australia from Jun 2019 to May 2020 are also shown in upper left corner.**

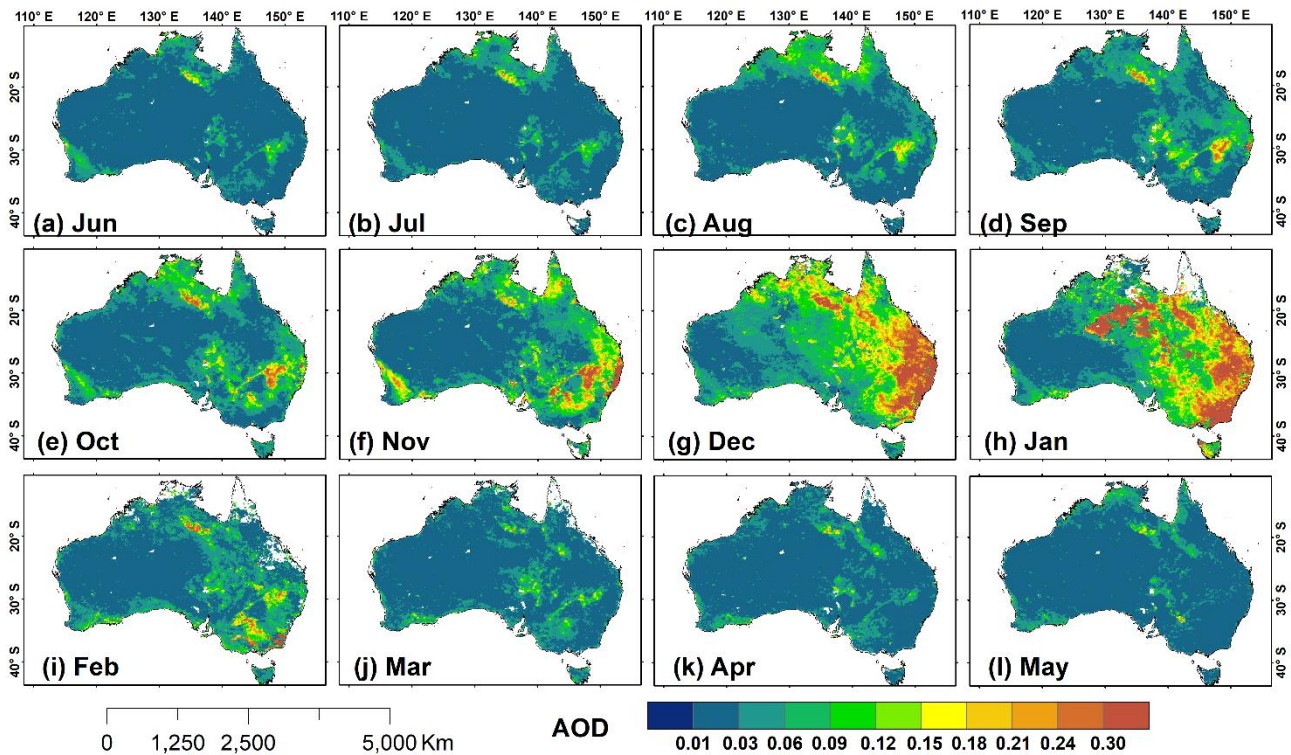

**Figure 11. Spatial distributions of monthly averaged Aqua MODIS DB AOD at 550 nm over Australia from Jun 2019 to May 2020.**

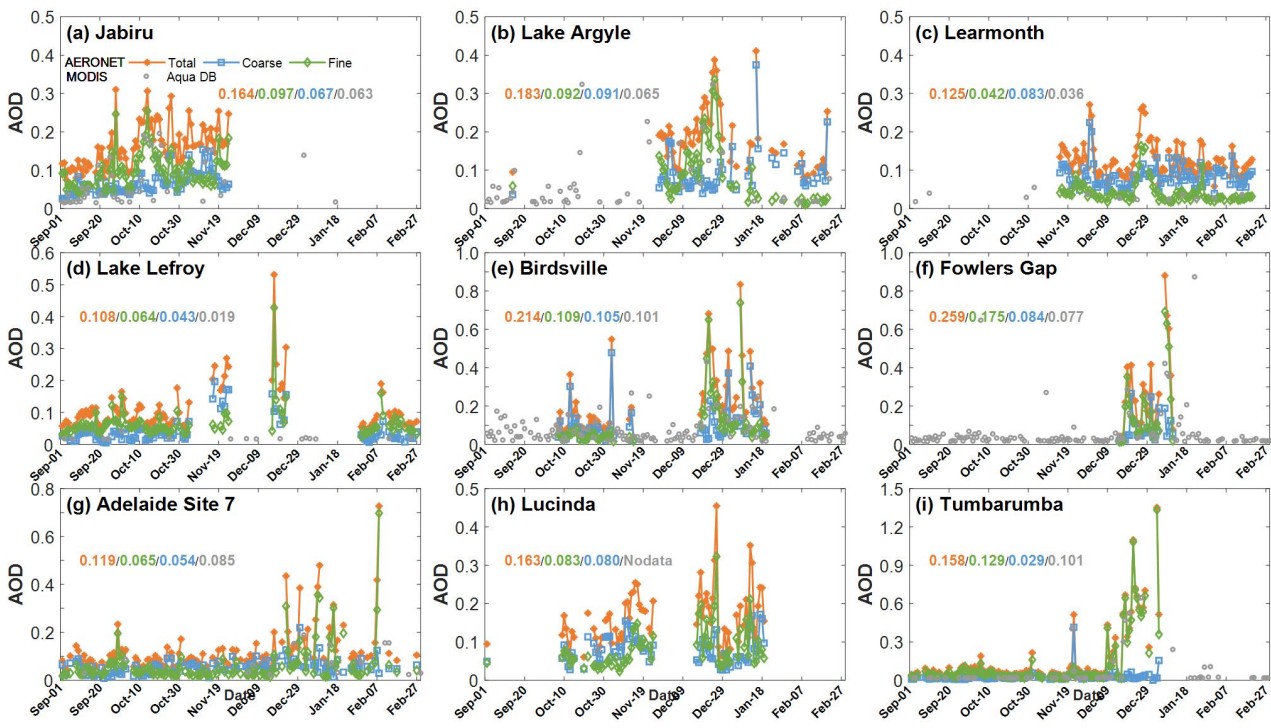

**Figure 12. Time series of average total, fine mode, and coarse mode AODs at 500 nm from AERONET, along with the Aqua MODIS DB AOD during September 2019-February 2020 at nine sites in Australia. The number above with different colors refers to the corresponding average values. Note: the data time period at each site is different.**

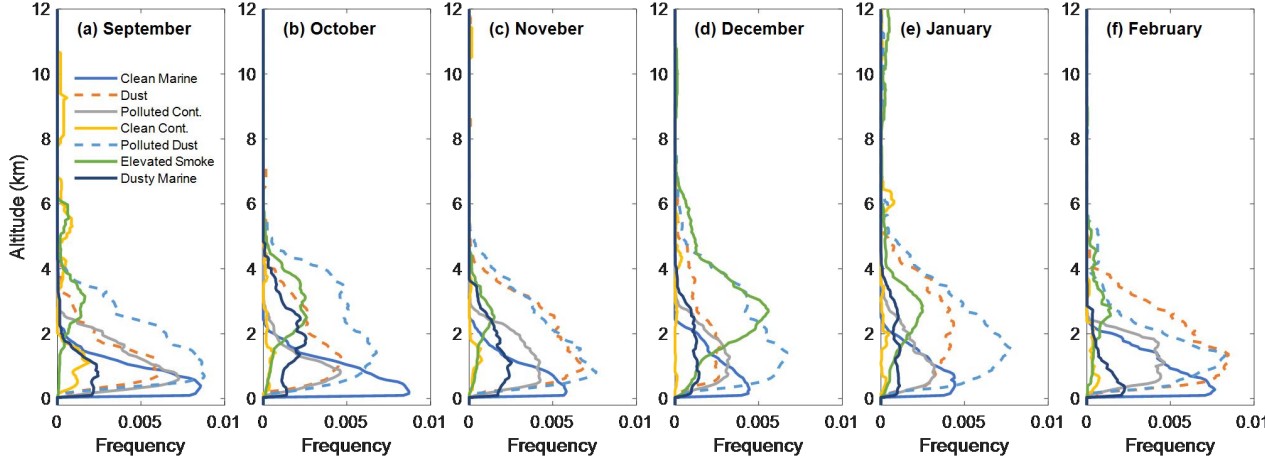

**Figure 13. Occurrence frequency profile of each aerosol type during the period September 2019-February 2020 from CALIPSO L3 aerosol profile data product in southeastern Australia (the selected domain is the gray shadow area in southeastern Australia in Fig.1(b)).**

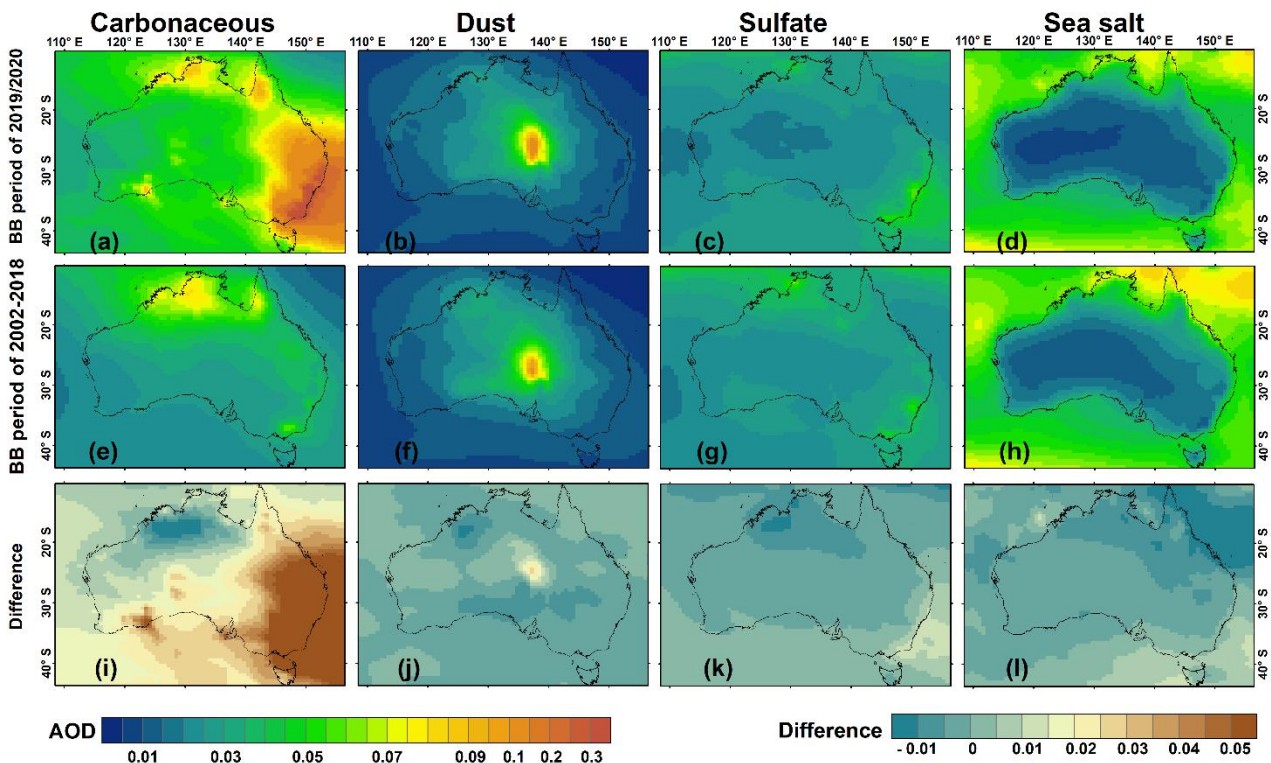

     **Figure 14. Spatial distributions of carbonaceous, dust, sulfate, and sea salt AODs from MERRA-2 during BB period of**
**2019/2020 (first row), BB period of 2002-2018 (second row), and the differences between the two periods (third row) over Australia.**

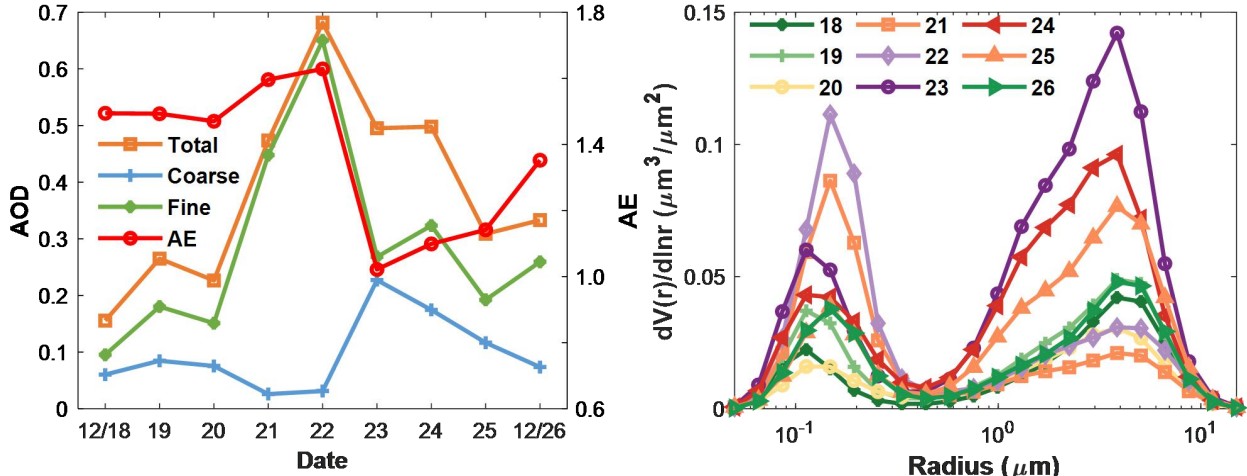

**Figure 15. Daily AOD and AE from AERONET during 18 December–26 December 2019 at Birdsville.**

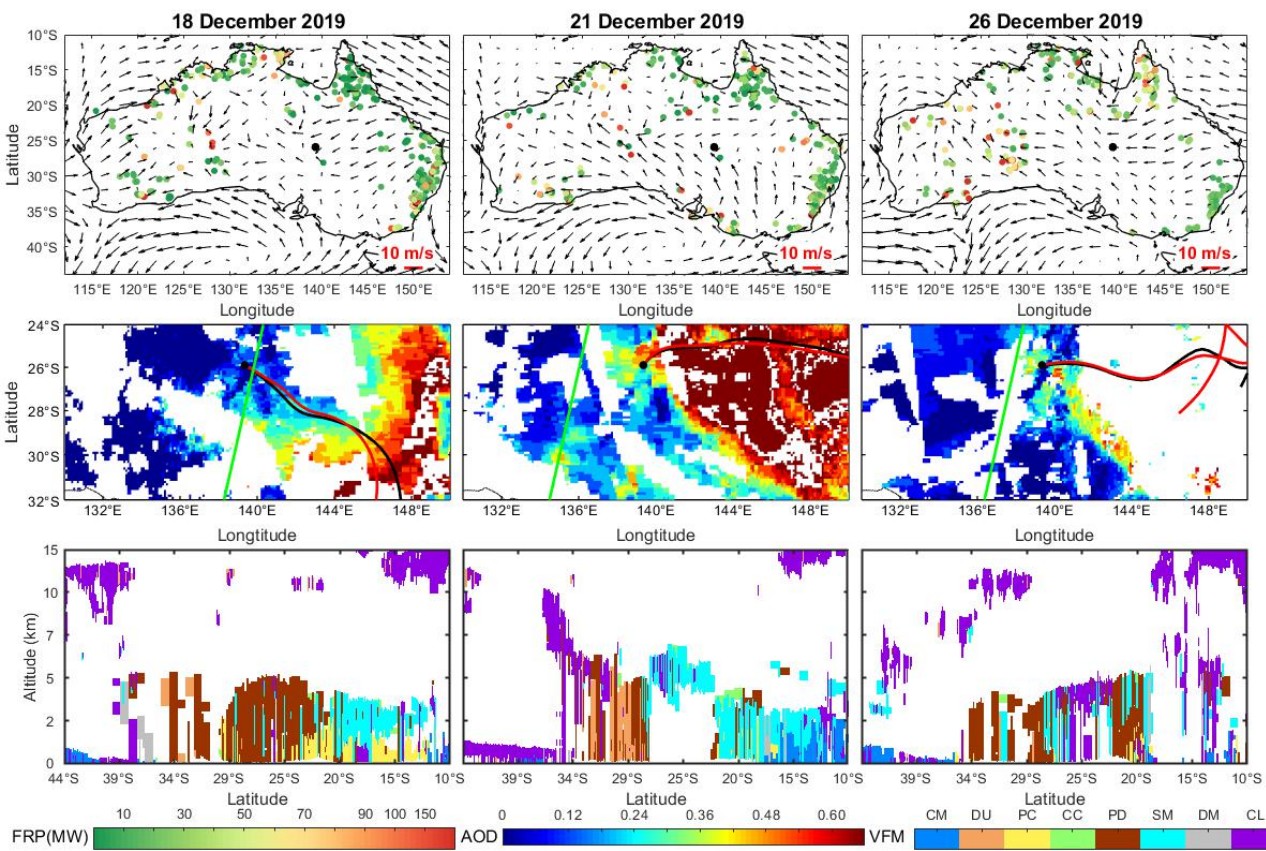

**Figure 16. Spatial distributions of fire spots (points) from MCD14ML and winds (arrows) from the ERA-5 over Australia (the first row), the Aqua MODIS DB AOD at 550 nm and 72 h back trajectories ending at two heights above the ground level (500 m in red and 200 m in black lines) at Birdsville (the second row), and the vertical feature mask of aerosol on 18, 21 and 26 December 2019 (the third row). (CM: Clean Marine; DU: Dust; PC: Polluted Cont.; CC: Clean Cont.; PD: Polluted Dust; SM: Smoke; DM: Dusty Marine; CL: Cloud). The black dot in in the panels of the first and second rows represent the Birdsville site. The green lines in the panels of the second row represent the scanning orbit path of the CALIPSO satellite.**

**Table 1. Site location and data time period at each site used in this study.**

| Site | Longitude | Latitude | Time span |
| --- | --- | --- | --- |
| Adelaide_Site_7 | 138.66 | -34.73 | 2006-2007;2017-2020 |
| Birdsville | 139.35 | -25.9 | 2005-2020 |
| Canberra | 149.11 | -35.27 | 2003-2017 |
| Fowlers_Gap | 141.70 | -31.09 | 2013-2020 |
| Jabiru | 132.89 | -12.66 | 2001-2007;2009-2020 |
| Lake_Argyle | 128.75 | -16.11 | 2001-2020 |
| Lake_Lefroy | 121.71 | -31.26 | 2012-2020 |
| Learmonth | 114.10 | -22.24 | 2017-2020 |
| Lucinda | 146.39 | -18.52 | 2009-2010;2013-2020 |
| Tumbarumba | 147.95 | -35.71 | 2019-2020 |