# Peer review of "Statistical aerosol properties associated with fire events from 2002 to 2019 along with a case analysis in 2019 over Australia"

_Atmospheric Chemistry and Physics, 2020_

## Referee Comment (RC1) · Anonymous Referee #1 · 3 Dec 2020

General comments: The manuscript "Statistical aerosol properties associated with fire events from 2002 to 2019 along with a case analysis in 2019 over Australia" discusses the contribution of biomass burning aerosols over Australia. Combining observation data and a trajectory model, it was found there was a significant difference of aerosol properties between biomass burning period and non-biomass burning period, and the impact of biomass burning aerosol is important. The data and methods used in this study are effective, the conclusions are partially reliable. However, there are still some concerns that need to be addressed.

Specific comments: 1. In this study, both ERA5 and MERRA-2 reanalysis datasets

are employed. Why not use the same dataset for meteorological variables and aerosol reanalysis. For example, MERRA-2 reanalysis dataset includes wind filed also, and ECMWF also provides aerosol reanalysis data (Copernicus Atmosphere Monitoring Service, CAMS). 2. Page 5 Line 179-180, how to get the conclusion of "similar emission from fires and regional transport of biomass burning aerosol in Australian continent" from the correlation between AOD and FRP, AOD and fire counts? 3. As shown in Fig. 8, the contribution of sea salt aerosol to the Australian continent is relatively small. However, the authors concluded that "The contributions of carbonaceous, dust, sulfate, and sea salt aerosols to the total aerosols were 26.24%, 23.38%, 26.36%, and 24.02% over Australia". So, maybe the contribution of sea salt aerosols to the Australian continent was overestimated by calculating the aerosol proportion in the whole region. 4. Page 9 Line 360-361, the authors mentioned that the relatively high coarse mode aerosol volume concentrations in southeastern Australia was mostly related to the fire-induced dust emissions caused by the pyro-convection during extreme fire events. What is the evidence for this conclusion? 5. Page 4 Line 147, the authors mentioned the use of precipitation from ERA-5. It should be described in Section 2.2.3. Additionally, what is the purpose of using total precipitation data? Why not use satellite, station or grid data? 6. Page 1 Line 21, "Carbonaceous" should be "carbonaceous". 7. Page 2 Line 45, "biomass burning aerosol" should be "biomass burning aerosols". 8. Page 3 Line 106-107, "level" in "level 2.0" and "Level 1.5" should be unified as "Level". 9. Page 3 Line 114-115, the full names of "DB" and "DT" are needed. 10. Page 6 Line 240 and Page 7 Line 257, is there some potential relation between the wildfire and dust events? 11. Page 7 Line 271, "MERRA" should be "MERRA-2".

---

## Referee Comment (RC2) · Anonymous Referee #2 · 7 Dec 2020

The Australian fires have recently drawn lots of attention, especially the three-month long and gigantic one last year. The manuscript presents a topical research to characterize the aerosol properties during the fire season in Australia, with a focus of the 2019-2020 extreme case. The authors' efforts in collecting and analyzing several observational and modeling data from multiple sources (e.g. MODIS/CALIPSO satellites, ground-based AERONET, MERRA2 reanalysis) are commendable. The manuscript is easy to follow and fit to the scope of ACP very well. I recommend its publication with ACP, while I also have minor comments below for the authors to address.

1. The authors have done a good job in analyzing the aerosol distribution in both spatial and temporal domain. As a step forward, it would be interesting to examine the spatiotemporal variations of the absorbing capability of the aerosols from the wildfires in Australia. Since the AERONET has single scattering albedo (SSA) as a product, it would be a low-hanging fruit to analyze it. In particular, how well the BC and OC from MERRA2 are correlated with the observed SSA is of great interest.

2. Please label out years in Figure 2.

3. How are the aerosol volume size distributions measured by the AERONET? If they are from remote sensing retrievals, how reliable they are? Some references are needed here.

4. About the comparison of AOD between MERRA2 and AERONET, I assume AERONET only report AOD in the non-cloudy days, while MERRA2 can calculate AOD anytime. Such a sampling issue needs to be mentioned.

5. Fig. 8, please clarify at which level the mass concentrations are. Near surface?

6. MERRA2-Aero has black carbon and organic carbon separately. Does the term "carbonaceous" in the paper refers to the summation of those two species? Please clarify.

7. What is the source of dust detected at heights from 2 to 5 km in November 2019, January, and February 2020? From fire or from the desert in the west?

---

## Author Comment (AC1) · 3 Feb 2021

We appreciate the reviewer's comments on the manuscript. All comments are highly valuable and helpful for us to improve our manuscript. We have studied them carefully and have addressed them in the revised manuscript. Attached file provides our point-by point responses to the reviewer's comments.

Please also note the supplement to this comment:
https://acp.copernicus.org/preprints/acp-2020-1139/acp-2020-1139-AC1-
supplement.pdf

[Figure]

**Supplement:**

**Reply to Anonymous Reviewer #1:**

**We appreciate the reviewer's comments on the manuscript. All comments are highly valuable and helpful for us to improve our manuscript. We have studied them carefully and have addressed them in the revised manuscript. Below are point-by point responses to the reviewer's comments.**

**Comments from the reviewer:**

1. In this study, both ERA5 and MERRA-2 reanalysis datasets are employed. Why not use the same dataset for meteorological variables and aerosol reanalysis. For example, MERRA-2 reanalysis dataset includes wind filed also, and ECMWF also provides aerosol reanalysis data (Copernicus Atmosphere Monitoring Service, CAMS).

   The reviewer proposed a good question. Indeed, both ERA5 and MERRA-2 reanalysis datasets include meteorological variables and aerosol data. There were two reasons to use meteorological variables from ERA-5. First, ERA-5 with a comparatively high spatial resolution (0.25°×0.25°) compared to the MERRA-2 (0.625°×0. 5°). Second, so far, there were some studies evaluating the applicability of this dataset. Bénédicte Jourdier evaluated the wind speed from ERA-5 and MERRA-2 over France and found that the MERRA-2 had large biases and overestimate wind speeds. Gruber et al. (2020) found that the wind power generation from ERA-5 outperformed MERRA-2 in the US, Brazil, South-Africa, and New Zealand. Cionni et al. (2018) showed that ERA-5 performs better than MERRA-2 in assimilating the precipitation during the period 2000-2017 in global scale. Therefore, in this study, we used the meteorological variables from ERA-5. In addition, the CAMS reanalysis is the latest global reanalysis data set of atmospheric composition produced by the Copernicus Atmosphere Monitoring Service, consisting of 3-dimensional time-consistent atmospheric composition fields, including aerosols, chemical species. However, the CAMS data currently covers the period 2003-2019. To obtain aerosol species data to analyze the long-term spatial and temporal characteristics of aerosols over Australia, and to study the change of aerosol properties during the 2019/2020 Australian Mega fire events, we used the MERRA-2 monthly data for the period 2001-2020.

Bénédicte Jourdier. Evaluation of ERA5, MERRA-2, COSMO-REA6, NEWA and AROME to simulate wind power production over France[J]. Advances in Science and Research, 2020, 17:63-77.

Gruber, Katharina; Regner, Peter; Wehrle, Sebastian; Zeyringer, Marianne; Schmidt, Johannes. Towards a global dynamic wind atlas: A multi-country validation of wind power simulation from MERRA-2 and ERA-5 reanalyses bias-corrected with the Global Wind Atlas[J]. arXiv:2012.05648v1. 2020.

Irene Cionni, Jaume Ramon, Llorenç Lledó, Harilaos Loukos, Thomas Noël. Validation of observational dataset and recommendations to the energy users.2018. https://s2s4e.eu/sites/default/files/2020-06/s2s4e_d31.pdf.

2. Page 5 Line 179-180, how to get the conclusion of "similar emission from fires and regional transport of biomass burning aerosol in Australian continent" from the correlation between AOD and FRP, AOD and fire counts?

The reviewer proposed a good question. We agree that It is not appropriate to get this conclusion. We have deleted the wrong description and modified our descriptions at Lines 184-190: "**The temporal AOD variations over Australia were well correlated with that of FRP (R=0.62) for the whole nineteen-year period, while the temporal AOD variations were weakly correlated with that of fire counts (R=0.43). Furthermore, the peak of the monthly mean AODs coincides with the FRP peak in the months of October-January each year (Table.S1). The correlation between AOD and fire counts was much higher (0.63-0.85) during the period of October-January than other months (-0.08-0.47). This was related to the intensive and frequent fires in the tropical savannas of northern Australia and the temperate southern Australia.**".

3. As shown in Fig. 8, the contribution of sea salt aerosol to the Australian continent is relatively small. However, the authors concluded that "The contributions of carbonaceous, dust, sulfate, and sea salt aerosols to the total aerosols were 26.24%, 23.38%, 26.36%, and 24.02% over Australia". So, maybe the contribution of sea salt aerosols to the Australian continent was overestimated by calculating the aerosol proportion in the whole region.

We appreciate this suggestion. We have added a figure (Figure 6) in the revised manuscript based on the suggestion from the reviewer#2. The Figure 8 now is Figure 9 in the revised manuscript. In this study, we conducted the conclusion by calculating the proportion based on the monthly averaged AOD of the four aerosol types (i.e. carbonaceous, dust, sulfate, and sea salt) over Australian continent from January 2001 to May 2020. It is really challenging for us to evaluate the aerosol types from MERRA-2 without sufficient information. However, we calculated the mean AOD of the four types of aerosols for the whole period, BB period, and Non-BB period over Australian continent, and found that the proportion of sea salt aerosol are 23.34%, 20.59%, 25,63% in whole period, BB period, Non-BB period, respectively (Table R1). The results from the two calculation methods do not differ significantly. We believe that this conclusion is relatively reliable, assuming that the four aerosol types from MERRA-2 are accurately estimated. In addition, roughly estimated from Figure 9, the sea salt contribution should also be around 20%.

Table R1 The mean values of four aerosol types from MERRA-2

| Period | Carbonaceous | Dust | Sulfate | Sea salt |
|---|---|---|---|---|
| Whole period | 0.266 | 0.209 | 0.231 | 0.215 |
| BB period | 0.392 | 0.251 | 0.271 | 0.237 |
| Non-BB period | 0.141 | 0.167 | 0.191 | 0.172 |

4. Page 9 Line 360-361, the authors mentioned that the relatively high coarse mode aerosol volume concentrations in southeastern Australia was mostly related to the fire-induced dust emissions caused by the pyro-convection during extreme fire events. What is the evidence for this conclusion?

We appreciate this suggestion. According to the result from CALIPSO, we found the occurrence frequency of dust increased at heights roughly from 2 to 5 km in November, January, and February. Wagner et al. (2018) indicated that fire radiative energy released by the combustion of the vegetation leads to a significant increase in near-surface wind speed, atmospheric turbulence, and vortices. Moreover, the removal of vegetation during the burning process and the accompanied dehydration and modification of the soil could consequently enhance the dust mobilization and uplift potential, which finally influenced the concentration and the mean size of aerosol particles over the fire region. In addition, according to McGowan and Clark. (2008), the dust from the Lake Eyre Basin can potentially affect the southeastern Australia through the southeast dust transport corridor. Therefore, we simulated the backward trajectories at Tumbarumba, which is located at southeastern Australia, at 1000 m above ground level. The backward trajectories showed that the airflows are from the Lake Eyre Basin during the period September 2019- February 2020 (Fig.R1). Overall, the increase of the occurrence frequency of dust in November 2019, January 2020, and February 2020 may be the result of fire-induced dust emissions caused by the pyro-convection during extreme fire events and long range transport of dust from the Lake Eyre Basin. We add a description about this at Lines 392-401: "**Wagner et al. (2018) indicated that fire radiative energy released by the combustion of the vegetation leads to a significant increase in near-surface wind speed, atmospheric turbulence, and vortices. Moreover, the removal of vegetation during the burning process and the accompanied dehydration and modification of the soil could consequently enhance the dust mobilization and uplift potential, which finally influenced the concentration and the mean size of aerosol particles over the fire region. McGowan and Clark. (2008) showed that dust from the Lake Eyre Basin can potentially affect the southeastern Australia through the southeast dust transport corridor. Therefore, the increase in the occurrence frequency of dust also explained the relatively high coarse mode aerosol volume concentrations at the sites in southeastern Australia (Fig.5), which was a result of the fire-induced dust emissions caused by the pyro-convection during extreme fire events and long range transport of dust from the Lake Eyre Basin.**".

[Figure]

Fig.R1 The 72 h back trajectories ending at Tumbarumba at 1000 m above ground level. The gray lines represent the total back trajectories in each month. The orange lines represent the cluster back trajectories.

Wagner, R., Jähn, M., and Schepanski, K.: Wildfires as a source of airborne mineral dust – revisiting a conceptual model using large-eddy simulation (LES), Atmos. Chem. Phys., 18, 11863-11884, 10.5194/acp-18-11863-2018, 2018.

5. Page 4 Line 147, the authors mentioned the use of precipitation from ERA-5. It should be described in Section 2.2.3. Additionally, what is the purpose of using total precipitation data? Why not use satellite, station or grid data?

Thank you for your suggestion. We have described the precipitation in Section 2.2.3. The total precipitation is the accumulated liquid and frozen water, comprising rain and snow, that falls to the Earth's surface. It is the sum of large-scale precipitation and convective precipitation. Mitchell et al. (2013) showed that the rainfall in the Australian wet season had a strong influence on aerosols. Therefore, we used the total precipitation data in this study. The results also showed that the extra rainfall in wet season likely promoted extra growth in flammable grasses, which leads to a strong maximum in aerosol optical depth in next dry season. In general, the observation stations of precipitation are unevenly distributed and lack of the spatial continuity. The satellite observation data of precipitation (such as TRMM data) could serve as a good choice to use. However, considering that the winds data we used are from ERA-5, to keep the data source of winds and precipitation the same, we still chose the ERA-5 precipitation data.

Several studies have shown that the precipitation data from ERA-5 have good performance compared to the satellite data. An evaluation by Peña-Arancibia et al. (2013) regarding the reanalysis datasets, satellite products, and an ensemble of these datasets in Australian and Asian regions showed that the ERA-Interim performed better than other individual datasets across a range of metrics for the Australian region. Furthermore, ERA5 is also closer than ERA‑Interim to the Tropical Rainfall Measuring Mission (TRMM)/3B43 in its representation of the temporal variability of monthly precipitation (Hersbach et al., 2020). The ERA-5 global‑mean correlation with monthly‑mean Global Precipitation Climatology Project (GPCP) data is 77%. This indicated that ERA-5 had good performance in assimilating the precipitation in Australia. We have mentioned the use of precipitation from ERA-5 in the revised manuscript at lines 147-149:"**ERA-5 adds additional characteristics to ERA-Interim reanalysis, making it even richer in climate information (Albergel et al., 2018). In this study, monthly U-wind, V-wind, and total precipitation from ERA-5 dataset were used for meteorology analysis.**"

Peña-Arancibia, J. L., van Dijk, A. I. J. M., Renzullo, L. J., and Mulligan, M.: Evaluation of Precipitation Estimation Accuracy in Reanalyses, Satellite Products, and an Ensemble Method for Regions in Australia and South and East Asia, J. Hydrometeorol., 14, 1323–1333,

https://doi.org/10.1175/JHM-D-12-0132.1, 2013.

Hersbach, H, Bell, B, Berrisford, P, et al. The ERA5 global reanalysis. Q J R Meteorol Soc. 2020; 146: 1999– 2049. https://doi.org/10.1002/qj.3803

6.  Page 1 Line 21, "Carbonaceous" should be "carbonaceous".

    Corrected.

7.  Page 2 Line 45, "biomass burning aerosol" should be "biomass burning aerosols".

    Corrected.

8.  Page 3 Line 106-107, "level" in "level 2.0" and "Level 1.5" should be unified as "Level".

    We have changed "level 2.0" as "Level 2.0".

9.  Page 3 Line 114-115, the full names of "DB" and "DT" are needed.

    Thank you for pointing it out, the full name of "DB" and "DT" has been added in the revised manuscript.

    Lines 120-122: "**Sayer et al. (2014) found that the terrain is arid and bright over much of Australia, the MODIS Dark Target (DT) algorithm often retrieves small positive or negative AOD, while the Deep Blue (DB) algorithm can retrieve more available values with 85% of AOD retrievals falling within the expected error envelope in Oceania.**"

10. Page 6 Line 240 and Page 7 Line 257, is there some potential relation between the wildfire and dust events?

    This is a good question. The answer is yes. Previous studies showed that significant fractions of mineral dust found in smoke plumes originating from most likely raised by strong, turbulent fire related winds **(Wagner et al., 2018; Ravi et al.,2012)**. We made corresponding changes to Section 3.1.2 accordingly.

    Lines 250-260: "**The result indicated that both fine mode and coarse mode volume concentrations significantly increased during BB period. During the BB period, fires cause a temporary reduction in vegetation cover, which can increase biomass burning emissions which are primarily fine aerosol particles. SSA also showed decreasing trends with increase in wavelengths in most months at Jabiru and Lake Argyle in northwestern Australia, especially during the BB period (Figs. 6a, 6b), which showed stronger absorption in the near-infrared bands. Fires also accelerate soil erosion by winds and promote dust emissions (Ravi et al.,2012). The coarse particles such as dust in northern Australia could have been entrained into the biomass burning plume from local soil and also been transported from central Australian deserts (Winton et al., 2016; Yang et al., 2020b). SSA values at Lake Argyle were lower (< 0.90 at 440 nm wavelength) during September-November, suggesting the relative dominance of absorbing aerosols such as dust and black carbon. Furthermore, the sea salt aerosols from ocean would also contribute to the differences in volume size distributions and SSA among various sites, or between BB and non-BB periods.**"

    Lines 282-286: "**The fine and coarse mode volume concentrations were both higher in December-January at Canberra during BB period (Fig. 5i). The increase in fine mode volume concentrations was attributed to the forest fires**

**in southeastern Australia, while the increase in coarse mode volume concentrations was mostly related to dust particles from forest fires and transported from central Australian deserts, along with sea salt particles from ocean (McGowan and Clark., 2008; Murphy et al., 2018; Yang et al., 2020b)."**

Wagner, R., Jähn, M., and Schepanski, K.: Wildfires as a source of airborne mineral dust – revisiting a conceptual model using large-eddy simulation (LES), Atmos. Chem. Phys., 18, 11863-11884, 10.5194/acp-18-11863-2018, 2018.

Winton, V. H. L., Edwards, R., Bowie, A. R., Keywood, M.,Williams, A. G., Chambers, S. D., Selleck, P. W., Desservettaz, M., Mallet, M. D., and Paton-Walsh, C.: Dry season aerosol iron solubility in tropical northern Australia, Atmos. Chem. Phys., 16, 12829–12848, https://doi.org/10.5194/acp-16-12829-2016, 2016.

McGowan, H., and Clark, A.: Identification of dust transport pathways from Lake Eyre, Australia using Hysplit, Atmospheric Environment, 42, 6915-6925, https://doi.org/10.1016/j.atmosenv.2008.05.053, 2008.

Murphy, B. P., Prior, L. D., Cochrane, M. A., Williamson, G. J., and Bowman, D. M. J. S.: Biomass consumption by surface fires across Earth's most fire-prone continent, Global Change Biology, 25, 2018.

Ravi, S., Baddock, M. C., Zobeck, T. M., and Hartman, J.: Field evidence for differences in post-fire aeolian transport related to vegetation type in semiarid grasslands, Aeolian Res., 7, 3–10,2012.

Yang, X., Zhao, C., and Yang, Y.: Long-term multi-source data analysis about the characteristics of aerosol optical properties and types over Australia, Atmos. Chem. Phys. Discuss., 2020, 1-50, 10.5194/acp-2020-921, 2020b.

11. Page 7 Line 271, "MERRA" should be "MERRA-2".
    Corrected.

---

## Author Comment (AC2) · 3 Feb 2021

We appreciate the reviewer's comments on the manuscript. All comments are highly valuable and helpful for us to improve our manuscript. We have studied them carefully and have addressed them in the revised manuscript. Attached file provides our point-by point responses to the reviewer's comments.

Please also note the supplement to this comment:
https://acp.copernicus.org/preprints/acp-2020-1139/acp-2020-1139-AC2-supplement.pdf

[Figure]

[Figure]

**Supplement:**

**Reply to Anonymous Reviewer #2:**

**We appreciate the reviewer's comments on the manuscript. All comments are highly valuable and helpful for us to improve our manuscript. We have studied them carefully and have addressed them in the revised manuscript. Below are point-by point responses to the reviewer's comments.**

**Comments from the reviewer:**

1. The authors have done a good job in analyzing the aerosol distribution in both spatial and temporal domain. As a step forward, it would be interesting to examine the spatiotemporal variations of the absorbing capability of the aerosols from the wildfires in Australia. Since the AERONET has single scattering albedo (SSA) as a product, it would be a low-hanging fruit to analyze it. In particular, how well the BC and OC from MERRA2 are correlated with the observed SSA is of great interest.

   We appreciate these valuable suggestions. Following these suggestions, we have performed the analysis of monthly averaged single scattering albedo (SSA) at the nine AERONET sites over Australia. The BC and OC from MERRA2 are correlated well with SSA from AERONET. For example, we found that SSA values at Lake Argyle were lower (< 0.90 at 440 nm wavelength) during September-November, suggesting the relative dominance of absorbing aerosols such as dust and black carbon. Furthermore, the northwestern Australia had always been the relatively high carbonaceous (BC+OC) AOD region during BB period, which is consistent with the assimilation from MERRA-2. In addition, we have added the discussions in our revised manuscript at several parts. For example,

   Lines 251-260: "**During the BB period, fires cause a temporary reduction in vegetation cover, which can increase biomass burning emissions which are primarily fine aerosol particles. SSA also showed decreasing trends with increase in wavelengths in most months at Jabiru and Lake Argyle in northwestern Australia, especially during the BB period (Figs. 6a, 6b), which showed stronger absorption in the near-infrared bands. Fires also accelerate soil erosion by winds and promote dust emissions (Ravi et al.,2012). The coarse particles such as dust in northern Australia could have been entrained into the biomass burning plume from local soil and also been transported from central Australian deserts (Winton et al., 2016; Yang et al., 2020b). SSA values at Lake Argyle were lower (< 0.90 at 440 nm wavelength) during September-November, suggesting the relative dominance of absorbing aerosols such as dust and black carbon. Furthermore, the sea salt aerosols from ocean would also contribute to the differences in volume size distributions and SSA among various sites, or between BB and non-BB periods.**".

   Lines 267-273: "**SSA showed an ambiguous wavelength dependence (i.e., increasing or decreasing with wavelengths) at Learmonth due to the presence of aerosol mixture (Fig. 5c). However, the average SSA values were less than 0.90 at 440 nm wavelength during late spring and summer at Learmonth, showing**

**absorbing properties of coarse particles, which was associated with the site's location in the North-Western dust pathway from the Australian interior deserts (e.g. the Gibson Desert and Great Victoria Desert). The average SSA values generally decreased with increasing spectral range at Lake Lefory possibly due to the anthropogenic emissions and biomass burnings (Yang et al., 2020) (Fig. 5d)."**

Line 279-281: **"In eastern Australia, the coarse mode aerosols were dominant in almost all seasons at Lucinda (Fig. 5h). The average SSA values generally increased with increasing spectral range with low values (<0.95) at Lucinda (Fig. 6h)."**.

[Figure]

**Figure 6. Monthly averaged single scattering albedo (SSA) at the nine AERONET sites over Australia. The warm-toned and cold-toned lines represent the aerosol size distributions in BB period and non-BB period, respectively. Note: Only SSA data from Tunbarumb is used due to the lack of SSA data at Canberra.**

2. Please label out years in Figure 2.

Thank you for your suggestion. We have revised the Figure 2 as suggested.

[Figure]

**Figure 2. Time series of monthly averaged fire radiative power (FRP), fire count from MCD14ML (a), and total precipitation from ERA-5 and Aqua MODIS DB AOD (b) in Australia.**

3. How are the aerosol volume size distributions measured by the AERONET? If they are from remote sensing retrievals, how reliable they are? Some references are needed here.

These are good questions and comments. The volume particle size distribution $dV(r)/dlnr$ ($\mu m^3$ /$\mu m^2$ ) is retrieved for 22 logarithmically equidistant discrete points ($r_i$) in the range of sizes 0.05 $\mu m \leq r \leq 15$ $\mu m$. The Cimel sky radiance measurements in the almucantar plane at 440, 675, 870, and 1020 nm (nominal wavelengths) in conjunction with the direct sun measured AOD at these same wavelengths were used to retrieve column-integrated aerosol size distributions ($dV(r)/dln(r)$ from 0.05 to 15 $\mu m$).

$$\frac{dV(r)}{dlnr} = V(r)\frac{dN(r)}{dlnr} = \frac{4}{3}\pi r^3 \frac{dN(r)}{dlnr}$$

We have added related references to describe the retrieval method and accuracy of aerosol volume size distributions from AERONET in the revised manuscript:

Lines 107-113: **"The retrievals of aerosol microphysical properties such as particle volume size distribution (dV(r)/dlnr) and single scattering albedo (SSA) are used in this study. The detailed retrieval algorithm can be found in Dubovik et al. (2000) and Dubovik et al. (2006), and hence are not reintroduced in this paper. The retrieval errors of dV(r)/dlnr did not exceed 10% in the maxima of the dV(r)/dlnr and may increase up to 35% for the minimum values of the dV(r)/dlnr in the intermediate particle size range (0.1 ≤ r ≤ 7 μm). The retrieval error of dV(r)/dlnr increased significantly for the edges of the particle size interval but did not significantly affect the derivation of the main features of the**

**particle size distribution (Dubovik et al., 2002).”**

Dubovik, O., Smirnov, A., Holben, B. N., King, M. D., Kaufman, Y. J., Eck, T. F., and Slutsker, I.: Accuracy assessments of aerosol optical properties retrieved from Aerosol Robotic Network (AERONET) Sun and sky radiance measurements, Journal of Geophysical Research: Atmospheres, 105, 9791-9806, 10.1029/2000jd900040, 2000.

Dubovik, O., Holben, B., Eck, T. F., Smirnov, A., Kaufman,Y. J., King, M. D., Tanré, D., and Slutsker, I.:Variability of Absorption and Optical Properties of Key Aerosol Types Observed in Worldwide Locations, J. Atmos. Sci., 59, 590–608,https://doi.org/10.1175/1520-0469(2002)059<0590:voaaop>2.0.co;2, 2002.

Dubovik, O., Sinyuk, A., Lapyonok, T., Holben, B. N., Mishchenko, M., Yang, P., Eck, T. F., Volten, H., Muñoz, O., Veihelmann, B., van der Zande, W. J., Leon, J.-F., Sorokin, M., and Slutsker, I.: Application of spheroid models to account for aerosol particle nonsphericity in remote sensing of desert dust, Journal of Geophysical Research: Atmospheres, 111, https://doi.org/10.1029/2005JD006619, 2006.

4. About the comparison of AOD between MERRA2 and AERONET, I assume AERONET only report AOD in the non-cloudy days, while MERRA2 can calculate AOD anytime. Such a sampling issue needs to be mentioned.

We agree with the reviewer and have made corresponding changes in the revised manuscript, which are in Lines 164-166: “**In this study, the quality controlled and cloud screened AOD from AERONET was used to evaluate the performance of MERRA-2 AOD for only cloud-free condition, while MERRA-2 AOD provides the aerosol information in both cloud-free and cloudy conditions.**”.

5. Fig. 8, please clarify at which level the mass concentrations are. Near surface?

We now have clarified the level of the mass concentrations which are surface mass concentrations, which are: “**Figure 8. Spatial distribution of carbonaceous, dust, sulfate, and sea salt mass concentrations near surface (a-h) and AOD (i-p) estimated by MERRA-2 during BB period and non-BB period over Australia.**”.

6. MERRA2-Aero has black carbon and organic carbon separately. Does the term “carbonaceous” in the paper refers to the summation of those two species? Please clarify.

We appreciate this suggestion. We have clarified the data source of carbonaceous, which are in Lines 309-311: “**Considering the similar emission sources of organic carbon and black carbon aerosols, carbonaceous aerosol is used in this study to refer to the summation of organic carbon and black carbon from MERRA-2.**”.

7. What is the source of dust detected at heights from 2 to 5 km in November 2019, January, and February 2020? From fire or from the desert in the west?

The reviewer proposed a good question. We simulated the backward trajectories at Tumbarumba, which is located at southeastern Australia, at 1000 m above ground level. The backward trajectories showed that the airflows were from the Lake Eyre Basin during the period September 2019- February 2020 (Fig.R1). In addition, Wagner et al. (2018) indicated that fire radiative energy released by the combustion of

the vegetation leads to a significant increase in near-surface wind speed, atmospheric turbulence, and vortices. Moreover, the removal of vegetation during the burning process and the accompanied dehydration and modification of the soil could consequently enhance the dust mobilization and uplift potential, which finally influenced the concentration and the mean size of aerosol particles over the fire region. Therefore, the increase of the occurrence frequency of dust in November 2019, January 2020, and February 2020 is most likely the combination result of fire-induced dust emissions caused by the pyro-convection during extreme fire events and long range transport of dust from the Lake Eyre Basin. Of course, it is challenging for us to quantify their relative contribution. We added a description about this at Lines 392-401: "**Wagner et al. (2018) indicated that fire radiative energy released by the combustion of the vegetation leads to a significant increase in near-surface wind speed, atmospheric turbulence, and vortices. Moreover, the removal of vegetation during the burning process and the accompanied dehydration and modification of the soil could consequently enhance the dust mobilization and uplift potential, which finally influenced the concentration and mean size of aerosol particles over the fire region. McGowan and Clark. (2008) showed that dust from the Lake Eyre Basin can also potentially affect the southeastern Australia through the southeast dust transport corridor. Therefore, the increase in the occurrence frequency of dust explained the relatively high coarse mode aerosol volume concentrations at the sites in southeastern Australia (Fig.5), which was most likely a combination result of the fire-induced dust emissions caused by the pyro-convection during extreme fire events and long range transport of dust from the Lake Eyre Basin.**".

[Figure]

Fig.R1 The 72 h back trajectories ending at Tumbarumba at 1000 m above ground level. The gray lines represent the total back trajectories in each month. The orange lines represent the cluster back trajectories.